# Photoinduced semiconductor-metal transition in ultrathin troilite FeS nanosheets to trigger efficient hydrogen evolution

Gang Zhou[1], Yun Shan[1,2], Longlu Wang[3], Youyou Hu[4], Junhong Guo[5], Fangren Hu[5], Jiancang Shen[1], Yu Gu[1], Jingteng Cui[2], Lizhe Liu[1] & Xinglong Wu[1]

The exploitation of the stable and earth-abundant electrocatalyst with high catalytic activity remains a significant challenge for hydrogen evolution reaction. Being different from complex nanostructuring, this work focuses on a simple and feasible way to improve hydrogen evolution reaction performance via manipulation of intrinsic physical properties of the material. Herein, we present an interesting semiconductor-metal transition in ultrathin troilite FeS nanosheets triggered by near infrared radiation at near room temperature for the first time. The photogenerated metal-phase FeS nanosheets demonstrate intrinsically high catalytic activity and fast carrier transfer for hydrogen evolution reaction, leading to an overpotential of 142 mV at 10 mA cm$^{-2}$ and a lower Tafel slope of 36.9 mV per decade. Our findings provide new inspirations for the steering of electron transfer and designing new-type catalysts.

---

[1] Key Laboratory of Modern Acoustics, MOE, Institute of Acoustics and Collaborative Innovation Center of Advanced Microstructures, National Laboratory of Solid State Microstructures, Nanjing University, Nanjing 210093, People's Republic of China. [2] Key Laboratory of Advanced Functional Materials of Nanjing, Nanjing Xiaozhuang University, Nanjing 211171, People's Republic of China. [3] School of Physics and Electronics, Hunan University, Changsha 410082, People's Republic of China. [4] Department of Physics, College of Science, Jiangsu University of Science and Technology, Zhenjiang 212003, People's Republic of China. [5] School of Optoelectronic Engineering and Grüenberg Research Centre, Nanjing University of Posts and Telecommunications, Nanjing 210023, People's Republic of China. These authors contributed equally: Gang Zhou, Yun Shan, Longlu Wang, Youyou Hu. Correspondence and requests for materials should be addressed to L.L. (email: lzliu@nju.edu.cn) or to X.W. (email: hkxlwu@nju.edu.cn)

Humankind has relied on fossil fuel in the past decades, causing significant degradation of global environment and frequent extreme weather. To solve this problem, huge efforts have been made to develop efficient and accessible energy conversion technologies in an attempt to produce sustainable and renewable energy sources[1–4]. Electrocatalytic water splitting for hydrogen evolution reaction (HER) is usually considered to be a promising and practical option because hydrogen can work as a versatile energy carrier, but is facing the dilemma of low conversion efficiency and high cost[5–7]. A grand challenge is the lack of inexpensive and excellent electrocatalysts with low overpotential and a small Tafel slope to drive hydrogen evolution[7–9]. So far, many earth-abundant materials such as transition metal dichalcogenides, sulfides, nitrides, and phosphides have been proposed to replace the costly Pt-based catalysts[10–16]. To achieve excellent HER performance, most research attentions have been directed toward preparation strategies, e.g., constructing various nanostructures[6,13] and introducing vacancies[9] or dopants[2,5]. It is generally known that a promising catalyst should possess high activity (appropriate Gibbs free energy), large number of exposed active sites (large superficial area), and low resistance for carrier transfer (good metal properties)[11,17,18]. These key elements strongly depend on the material's intrinsic physical properties. In principle, the design of excellent catalysts should be proceeded from the regulation of material's essence, such as structural phase transition[10,19–21]. If a semiconductor–metal transition is realized at near-room temperature via light exposure, the catalytic activity and carrier transfer will be efficiently regulated in a simple way.

Herein, we report the highly efficient hydrogen evolution triggered by photoinduced semiconductor–metal transition at near-room temperature for the first time. This phase transition occurs in ultrathin troilite FeS nanosheets vertically grown on carbon fiber cloth (CFC). The inexpensive troilite FeS material is chosen as the candidate since previous reports have confirmed that a semiconductor–metal transition easily occurs in FeS bulk material at ~400 K[22–24]. Our work witnesses the ultrathin FeS nanosheets with more active sites that have obviously enhanced the response to near-infrared (NIR) radiation, which can impel more light-excited carriers to join into the phase transition process and drastically lower the transition temperature near room temperature. The photogenerated metal-phase FeS nanosheets exhibit enhanced carrier transfer and higher catalytic activity, which is about eight times higher in HER performance than the semiconductor phase.

## Results

**Characterization of troilite FeS nanosheets**. The ultrathin FeS nanosheets were synthesized by a citric-assisted solvothermal procedure with ferrous chloride and thiourea as the Fe and S sources, respectively. As for hydrogen evolution, the ultrathin FeS nanosheets were vertically aligned onto the CFC substrate to serve as a functional electrode (Supplementary Figs. 1 and 2). The scanning electron microscopy (SEM) image in Fig. 1a shows that the FeS nanosheets on CFC are vertically aligned with three-dimensional porous textures. The FeS nanosheets are observed to

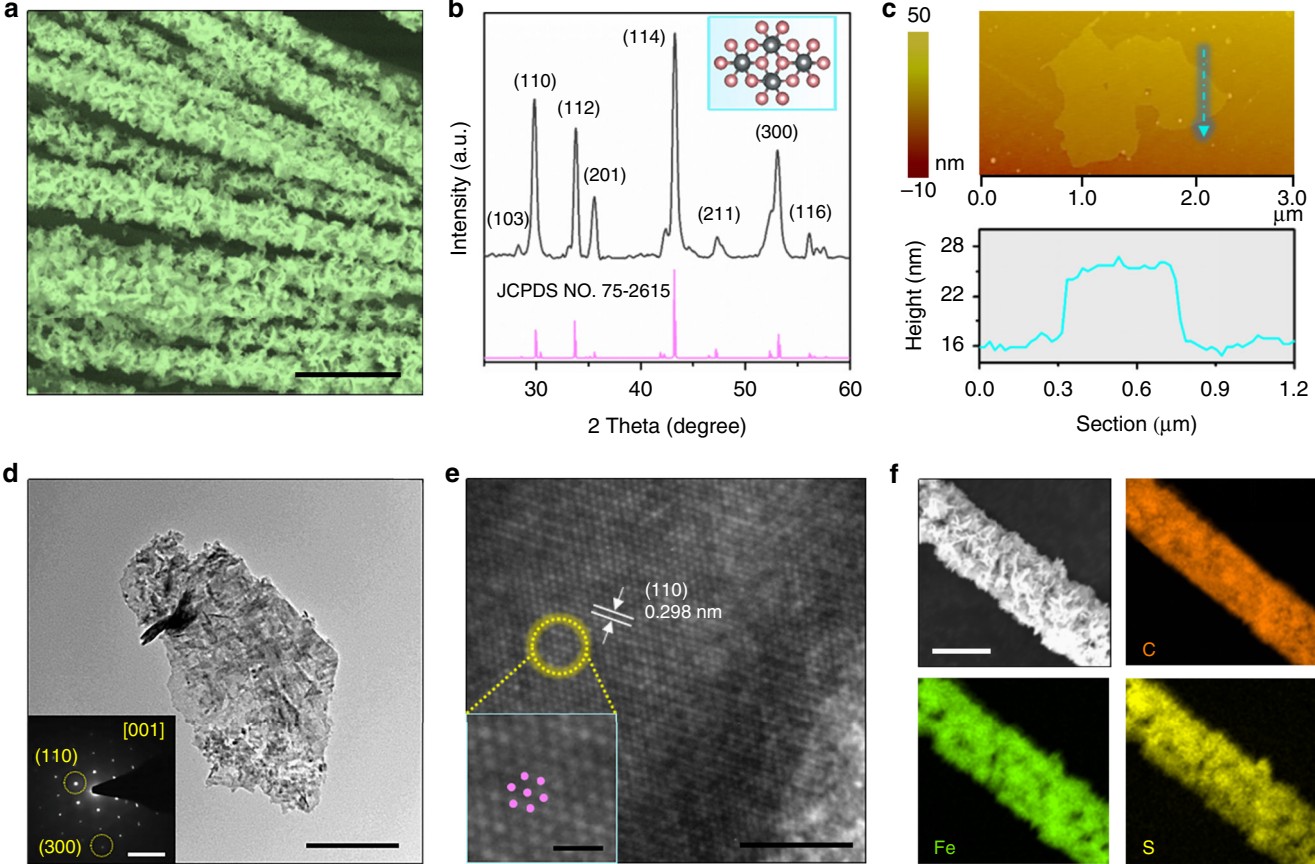

**Fig. 1** Characterization of ultrathin troilite FeS nanosheets. **a** FE-SEM image of FeS nanosheets grown on CFC (scale bar, 50 μm). **b** XRD patterns of FeS nanosheets. Inset: atomic structural model. **c** Top panel: AFM topographical image of a FeS nanosheet. Bottom panel: height profile of a FeS nanosheet is marked by arrow line. **d** TEM image of a FeS nanosheet (scale bar, 0.4 μm). Inset: the SAED pattern. **e** HR-TEM image of a FeS nanosheet (scale bar, 5 nm). Inset: the enlarged image (scale bar, 1 nm). **f** EDS mapping of FeS nanosheets grown on one carbon fiber (scale bar, 10 μm)

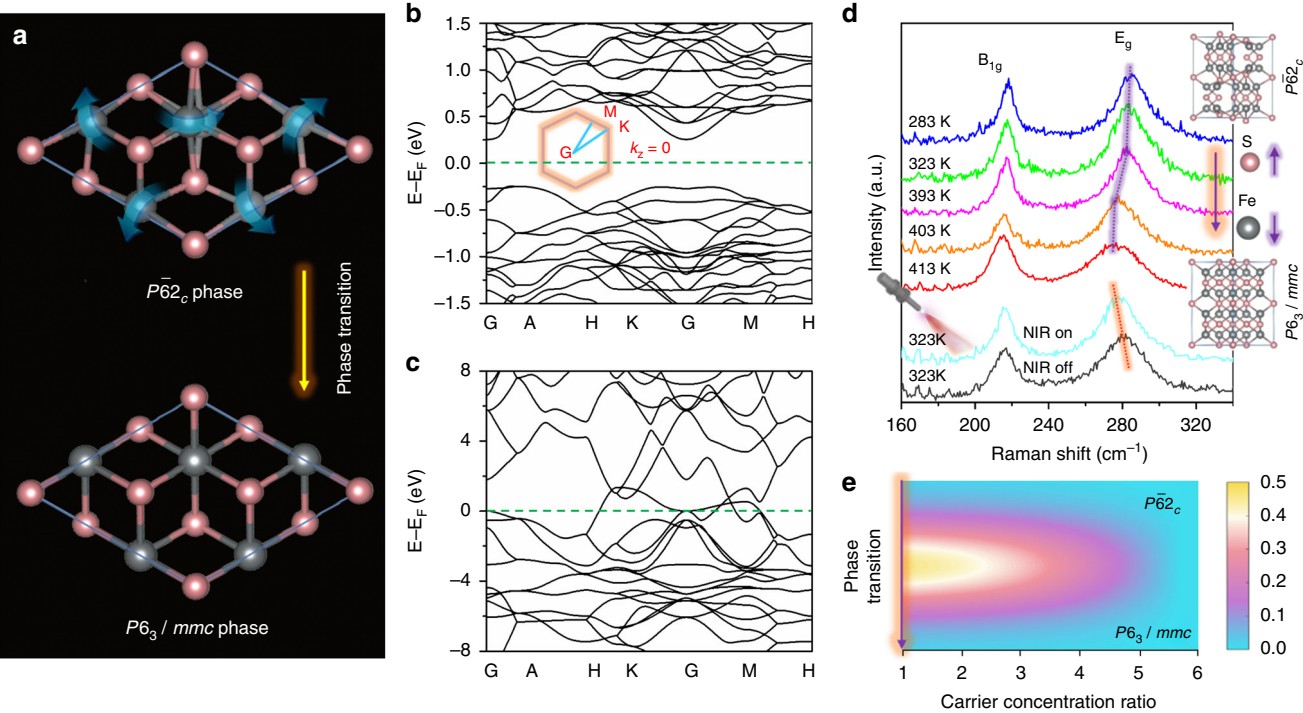

**Fig. 2** Phase transition of FeS nanosheets. **a** Structural transition from distorted $P\bar{6}2_c$ phase to high-symmetry $P6_3/mmc$ phase. The displacement vectors are marked by blue arrows. **b** The band structure of $P\bar{6}2_c$ phase. **c** The band structure of high-symmetry $P6_3/mmc$ phase. **d** In situ Raman spectra as functions of temperature with or without NIR light irradiation. **e** The potential barrier of phase transition as a function of carrier concentration ratio

be well-organized and connected with each other to form a stereoscopic structure with a large fraction of exposed edges and patulous pores. The average heights of dense loading of vertical FeS nanosheets are 700–800 nm beneficial for light response (Supplementary Fig. 3). The X-ray diffraction (XRD) pattern in Fig. 1b and Supplementary Fig. 4 shows that diffraction peaks are undoubtedly identical to that of troilite FeS with $P\bar{6}2_c$ symmetry phase (JCPDS NO. 75–2615). The X-ray photoelectron spectrum (XPS in Supplementary Fig. 5) further evidences the element composition and the successful preparation of FeS nanosheets. The atomic force microscopy (AFM) (Fig. 1c) and transmission electron microscopy (TEM) (Fig. 1d) images of the nanosheets exfoliated from CFC reveal that the flower-like nanosheets are of 10 nm in thickness and 1.2 μm in width. The high-resolution TEM (HR-TEM) images in Fig. 1e and selected area diffraction pattern (in the inset of Fig. 1d) confirm that high-crystallinity FeS with a resolved 0.298 nm lattice fringe corresponding to (110) planes is synthesized successfully. The enlarged HR-TEM image of a typical crystalline region shown in the inset in Fig. 1e displays the distorted hexagonal configuration of atoms, which is in consistence with the structure of troilite in the inset of Fig. 1b. In addition, energy-dispersive X-ray spectrum (EDS, Supplementary Fig. 6) and elemental mapping in Fig. 1f evidence that the Fe and S elements with a ratio of 1:1 are uniformly distributed in the structure.

**Verification of semiconductor–metal-phase transition in FeS nanosheets**. Previous reports have confirmed that a structural phase transition in FeS bulk material occurs at 400 K[22–24], changing the crystal symmetry from troilite structure with space group $P\bar{6}2_c$ to the so-called NiAs-type structure with high-symmetry hexagonal space group $P6_3/mmc$. To display this phase transition more explicitly, the structural distortion of turning $P\bar{6}2_c$ into $P6_3/mmc$ is shown in Fig. 2a. As the distorted Fe atoms relax to hexagonal sites, an indirect band gap of ~0.5 eV

degenerates in the electronic structure marking a semiconductor-to-metal transition, corresponding to Fig. 2b and Fig. 2c, respectively (Supplementary Fig. 7). To verify the phase transition in our sample, the in situ Raman spectra as functions of temperature are displayed in Fig. 2d. The characteristic peaks at ~285 $cm^{-1}$ and ~218 $cm^{-1}$ correspond to the out-of-plane $E_g$ mode and the in-plane $B_{1g}$ mode, which are attributed to the opposite vibration of the S atom with respect to the Fe atom perpendicular (as shown in the inset) and parallel to the plane[25], respectively. The two peaks respond differently to the temperature. The $B_{1g}$ peak remains unchanged at 218 $cm^{-1}$, but the $E_g$ mode slightly downshifts from 285 $cm^{-1}$ (283 K) to 281 $cm^{-1}$ (393 K), then rapidly shifts to 277 $cm^{-1}$ (403 K), and keeps near constant at ~413 K as the temperature increases. This indicates the out-of-plane crystallization breakage induced by phase transition at ca. 403 K (detailed change in Raman peak position, see Supplementary Fig. 8a). Besides, two peaks at 397 K and 386 K in the heating and cooling runs are observed in the curve of a differential scanning calorimeter (DSC) (Supplementary Fig. 8b), respectively, confirming that the semiconductor–metal transition can occur at ~397 K, consistent with Raman spectrum measurements (Supplementary Fig. 8a). Note that the phase transition temperature of ultrathin FeS nanosheets decreases to 323 K under NIR light (1064 nm) radiation because the $E_g$ mode under NIR light radiation at 323 K downshifts to the same position of that at 403 K without NIR light, as shown in the lower panel of Fig. 2d and Supplementary Fig. 8. And, this decrease in phase transition temperature cannot happen under visible light radiation that causes no change for the out-of-plane $E_g$ mode (Supplementary Fig. 9). The change in Raman spectra induced by NIR light radiation implies the semiconductor–metal-phase transition, which is consistent with the $^{57}Fe$ MÖssbauer spectra (Supplementary Fig. 10 and Supplementary Note 1).

The dependence of the phase transition potential barrier on the photogenerated carrier concentration in Fig. 2e discloses that the

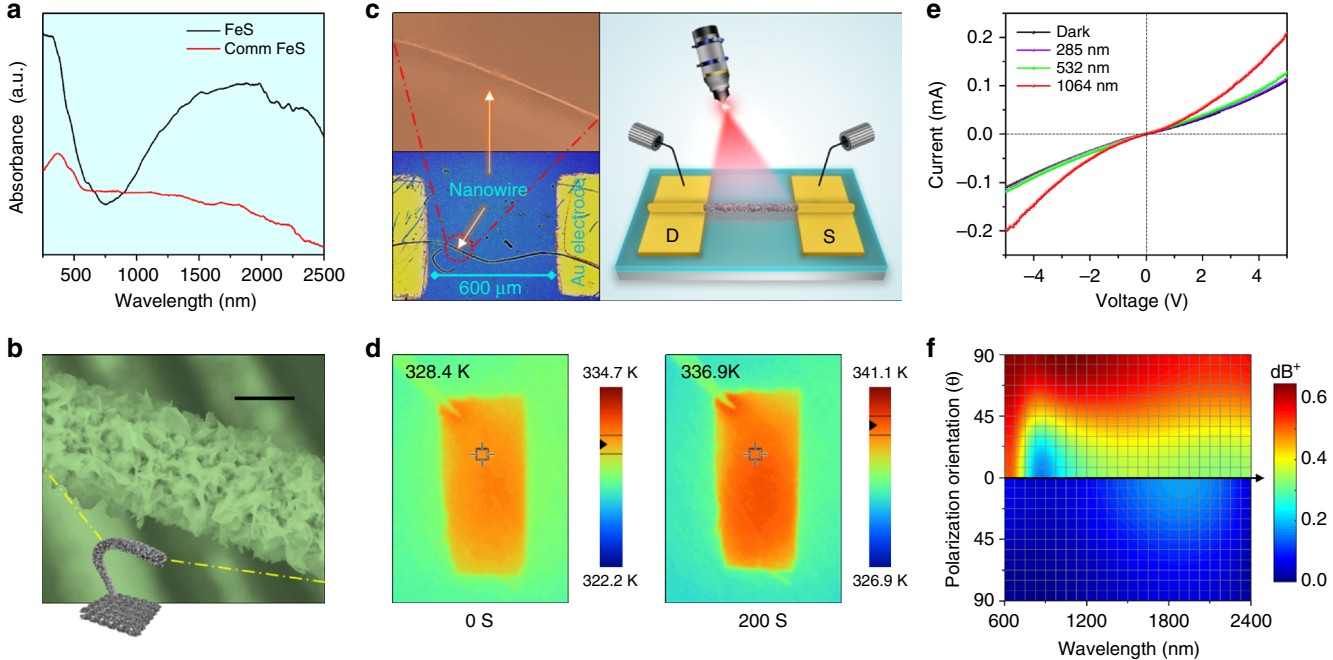

**Fig. 3** Physical mechanism of photoinduced phase transition. **a** Absorption spectra of ultrathin FeS nanosheets and commercial FeS powders. **b** FE-SEM image of a single carbon fiber covered by ultrathin FeS nanosheets (scale bar, 5 μm). **c** Schematic and optical image of a single photodetector structure. **d** Infrared imaging pictures of FeS/CFC. **e** I–V curves of the photodetector without and with light illumination at different wavelengths. **f** Absorption behavior as a function of polarization orientation and incident light wavelength

potential barrier decreases sharply from 0.46 (yellow) to 0.0 eV (cyan) as the carrier concentration increases by 5.7 times. This phase transition mechanism involving photogenerated carriers is schematically displayed in Supplementary Fig. 11. The absorption spectra in Fig. 3a show that absorption by the FeS nanosheets in the infrared range is higher than that of commercial FeS powders with a larger irregular size of 50–150 μm (Supplementary Fig. 12), and this has an advantage in generating photogenerated carriers. To further discern the contribution of NIR-generated carriers to the phase transition, the single carbon fiber sufficiently covered by ultrathin FeS nanosheets was extracted from carbon cloth (Fig. 3b) and fabricated into a photodetector. As shown in the left panel of Fig. 3c, two Au electrodes with a separation of 600 μm are deposited on the SiO$_2$/Si substrate to serve as the source and drain, respectively. The photodetector is exposed to different light to investigate its optical response, as shown in the right panel of Fig. 3c. The current vs. voltage (I–V) curves acquired at 328.4 K without (dark) and with 285- and 532-nm light radiation disclose that they all display linear and symmetrical behavior, indicating good Ohmic contact between the electrodes and FeS nanosheets (Fig. 3e). In contrast, the photocurrents not only increase drastically but also display nonlinear behavior under NIR light (1064 nm) radiation, which can be attributed to electronic conductivity variation induced by semiconductor–metal transition. This is in line with the Raman spectra in Fig. 2d. Based on the above photoelectric response, the concentration of a photogenerated carrier is increased from $7.79 \times 10^{12}$/cm$^3$ (semiconductor phase) to $9.28 \times 10^{13}$/cm$^3$ (metal phase) (~10 times), which can rapidly reduce the value of phase transition potential barrier. This value could meet the requirement of phase transition according to our theoretical simulation in Fig. 2e, which can be evaluated from the equation as follows:

$$\Delta I_p = q * U * \left( \Delta n \mu_n + \Delta p \mu_p \right) / s \qquad (1)$$

where $\Delta n = \Delta p$ is the number of photogenerated electron–hole pairs, $\mu_n$ and $\mu_p$ the electron and hole mobility (Supplementary Figs. 13–14), $U$ is the applied electric field, and $s$ is the cross-section area for current flow.

The infrared imaging pictures of FeS/CFC in Fig. 3d indicate that the temperature (328.4 K) is only enhanced by 8.5 K after 200 s of NIR light radiation. Thus, the photothermal contribution can be ruled out because our measurement only lasts around 30 s and the slight change in temperature cannot lead to a rapid increase in the photocurrent (Supplementary Fig. 15). To confirm this obvious optoelectronic behavior, the relation between absorption and polarization orientation of incident light is simulated by the FDTD method (Supplementary Fig. 16), and shown in Fig. 3f. Maxwell's electromagnetic theory holds that the electrical vector of incident light can be decomposed into P-wave (perpendicular to the material's surface) and S-wave (parallel to the material's surface) components, in which the P-wave component is mainly responsible for absorption. According to the theory of Fresnel's law[26], in the structure with vertically aligned nanosheets, more incident lights are expressed as P-wave polarization, leading to higher absorption capacity, as shown in the upper panel in Fig. 3f. Additionally, the hierarchical and porous structure of nanosheets can reduce surface reflection and increase internal light scattering and multi-reflection (Supplementary Fig. 17), resulting in a longer optical path length for light transport to enhance light absorption[27]. Therefore, compared with covered FeS without a vertically aligned configuration in the bottom panel of Fig. 3f, our sample has a remarkably enhanced absorption. These experimental and theoretical results evidence that excellent optical absorption and photogenerated carriers in our sample make semiconductor–metal transition occur at near-room temperature.

**HER performance**. The density functional theory (DFT) calculations (see Methods) are carried out to evaluate the HER activity

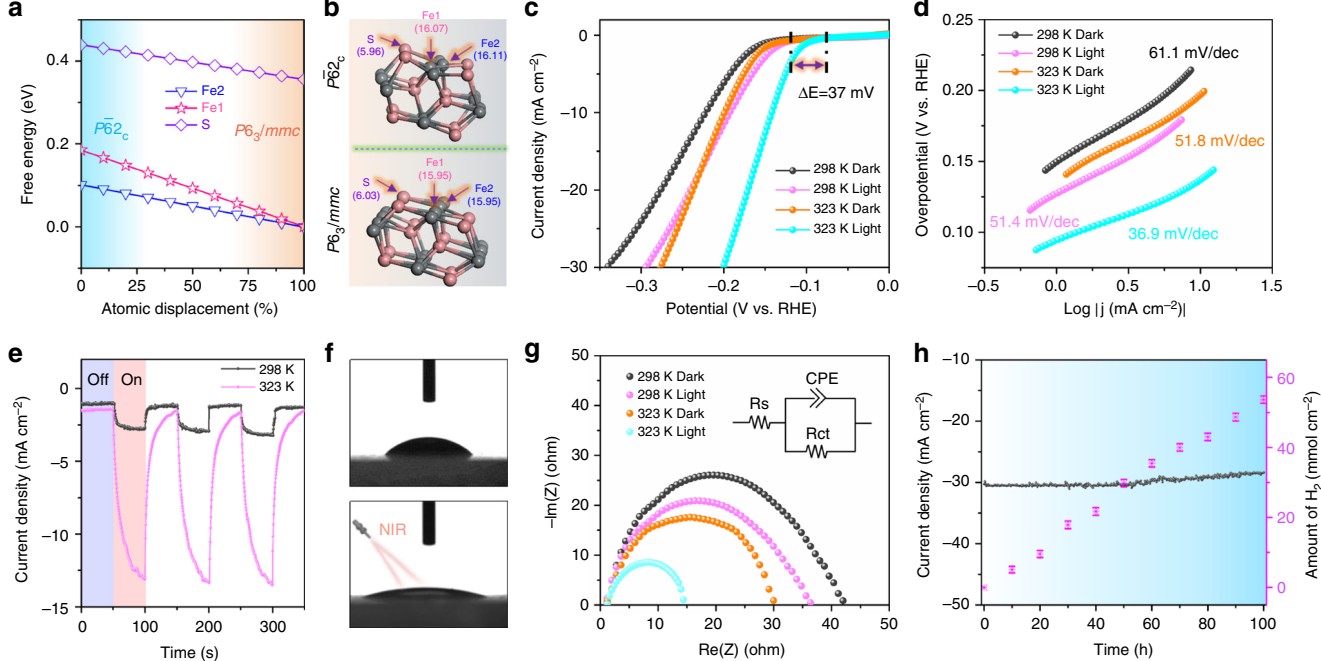

**Fig. 4** Practical application of HER. **a** The Gibbs free energy as a function of the fraction of the atomic distortion between the $P\bar{6}2_c$ (0%) and $P6_3/mmc$ (100%) phase. **b** Atomic structural models of FeS (100) surface for $P6_3/mmc$ phase (bottom panel) and its corresponding surface in $P\bar{6}2_c$ phase (top panel), in which the Mulliken charges are also marked. **c** Polarization curves and **d** Tafel plots of the FeS/CFC at different temperatures with or without NIR light irradiation. **e** Amperometric $I$–$t$ curves under chopped NIR light irradiation at different temperatures. **f** The differences in hydrophilicity with or without NIR light irradiation using contact angle measurement. **g** EIS curves in dark and light fields at different temperatures with an equivalent circuit (inset). **h** Chronoamperometry curve (left) of the FeS/CFC at a constant potential of −0.2 V (vs. RHE) accompanied by the continuous hydrogen measurement (right)

of the superficial S and Fe sites[28], as shown in the upper panel in Fig. 4b. Here, the hydrogen adsorption free energy $\Delta G_H$ was determined for different amplitudes of phase transition (Fig. 4a). $\Delta G_H$ is known to scale with activation energies and the optimal value of $\Delta G_H$ is 0 eV, where hydrogen is bound neither too strongly nor too weakly[11,29,30]. With an increase in amplitude of the pattern of displacement that drives FeS from the semi-conductor $P\bar{6}2_c$ phase (0%) to metal $P6_3/mmc$ phase (100%) (the structural transition in Fig. 4b), the $\Delta G_H$ changes linearly and finally converges to ~0 eV (Fe1 and Fe2 site), which offers flex-ibility for tuning the activity and simultaneously carrier transfer (semiconductor–metal transition). The atomic density of state (DOS) analysis confirms that the highest occupied orbital is composed by not $p$-states of S atoms but $d$-states of Fe atoms, which is beneficial for optimizing the electronic structures to improve HER performance.

To demonstrate the contribution of light-induced semiconductor–metal-phase transition, Fig. 4c shows typical polarization curves of FeS/CFC electrodes under various control conditions. A simple rise in temperature, i.e., from 298 K to 323 K in the dark, only slightly increases the HER activity as thermodynamic perturbation can be negligible, consistent with our predication in Fig. 3. Under NIR light radiation correspond-ing to the semiconductor–metal-phase transition, the HER activity is significantly improved at 323 K, as evidenced by the onset potential (0.5 mA cm$^{-2}$) positively shifting to −94 mV. The corresponding Tafel plots show that the Tafel slope decreases from 61.1 to 51.8 mV dec$^{-1}$ when the temperature increases from 298 K to 323 K and meanwhile a similar drop from 61.1 to 51.4 mV dec$^{-1}$ in the Tafel slope occurs at a fixed temperature of 298 K with the introduction of NIR radiation (Fig. 4d) (detailed values in Supplementary Table 1). However, a smaller Tafel slop (36.9 mV dec$^{-1}$) is achieved during NIR radiation at 323 K,

indicating a chemical rearrangement triggered by phase transition before $H_2$ desorption. Moreover, the Tafel slope suggests that Volmer–Heyrovsky mechanism plays a predominant role in determining the HER rate and the electrochemical desorption step is the rate-limiting step[7,31,32]. From the intercept of the linear region of Tafel plot, the exchange current density[21,33] ($j_0$, the intrinsic electron transfer rate between the electrode and the electrolyte) of 3.1 μA cm$^{-2}$ is obtained for the FeS/CFC electrodes at 323 K under NIR radiation. The electrochemical active surface area (ECSA) in Supplementary Fig. 18 discloses that the $C_{dl}$ of $P6_3/mmc$ phase has the ECSA value of 552.5 cm$^2$, which is about 4.5 times larger than that of $P\bar{6}2_c$ phase (122.5 cm$^2$), suggesting the enhanced catalytic active sites via phase transition[17,34]. The HER performance evidences good electrocatalytic activities of FeS/CFC electrodes in comparison with that of other Fe-based materials (Supplementary Table 2).

The transient $I$–$t$ curves for FeS/CFC electrodes present the responses of the same electrode to light radiation at 298 K and 323 K, respectively (Fig. 4e). Upon NIR light radiation, the reproducible current at 323 K rapidly increases about six times larger than that at 298 K. This NIR-induced significant current enhancement and gradual current decay once in the dark can be assigned to phase transition because the contribution of temperature is too limited to be considered, in view of the results in Figs. 2d, 4d, and Supplementary Fig. 19. Besides, we also find that the FeS/CFC electrodes become more hydrophilic under NIR light radiation, which is good for HER enhancement[17] (Fig. 4f).

The electrochemical impedance spectra (EIS) of FeS/CFC electrodes in the dark and the NIR light field disclose that the conductivities are strongly dependent on NIR illumination, especially at 323 K (Fig. 4g). The remarkable reduction in charge transfer resistance ($R_{ct}$) induced by NIR light radiation (detailed values in Supplementary Table 3) further suggests that the

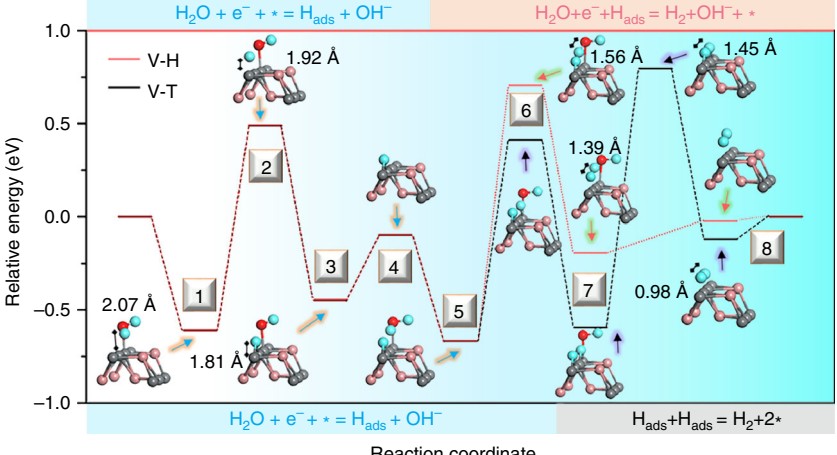

**Fig. 5** Schematic configuration-coordinate diagrams for the HER mechanism. Free energy vs. the reaction coordinates of HER for Volmer–Heyrovsky (V–H) and Volmer–Tafel (V–T) process. (Fe, gray spheres, S, pink spheres, H, blue spheres, O, red spheres)

semiconductor–metal transition not only provides higher catalytic activity but also facilitates the charge and mass transfer. Finally, the stability of the FeS/CFC electrodes was also tested to evaluate the electrode performance. The $I$–$t$ curve displays no significant change even after 100 h and the amount of hydrogen is proportional to the reaction time, indicating the constant hydrogen evolution (Fig. 4h). More importantly, the original morphology, the local crystal structure, and the chemical composition remain unaffected after the long-time stability testing (Supplementary Fig. 20).

**Proposed HER mechanism**. To elucidate the HER mechanism, the activation, adsorption, and reaction process of the $H_2O$ molecule onto the FeS (100) surface of $P6_3/mmc$ phase are investigated systematically via the climbing nudged elastic band method[35,36], and the corresponding reaction pathways along the Volmer–Heyrovsky (V–H) and the Volmer–Tafel (V–T) mechanisms are calculated and shown in Fig. 5, respectively. First, a single $H_2O$ molecule adsorbs onto the Fe site with a distance of 2.07 Å (S1), and then is slowly cracked into H* and OH* after overcoming a 1.18-eV barrier at S2 followed by formation of a stable Fe–H bond and Fe–OH bonds with an energy release of 0.94 eV (S3). Subsequently, the OH* is dissociated from the Fe atom (S4), and another $H_2O$ molecule is attracted to another neighboring Fe site (S5). Different from the V–T mechanism in the second step, the generated H* at the Fe site in the first step begins to combine with another H* in adsorbed $H_2O$*, in which the produced $H_2$* and OH* occupy two neighboring Fe sites (S7), respectively. To realize this process, an activation energy of 1.37 eV, including the cleavage of a HO–H bond in $H_2O$* and a Fe–H bond, needs to be climbed at the transition state (S6). Eventually, after overcoming a potential barrier from OH* release (S8), the molecular hydrogen with a stable configuration begins to dissociate from catalyst surface in the end. For the V–T mechanism, the second $H_2O$ molecule is adsorbed onto the next nearest neighboring Fe site to form another Fe–H bond, but the two adjacent H* atoms bridging a transition state with a distance of 1.45 Å by orbital hybridization need to overcome a corresponding higher activation energy of 1.39 eV.

## Discussion

In summary, we have demonstrated that the semiconductor–metal transition in ultrathin FeS nanosheets could occur under simple

NIR light radiation at 323 K to trigger highly efficient hydrogen evolution. This phase transition not only sharply enhances intrinsic catalytic activity but also obviously facilitates charge and mass transfer, producing an overpotential of 142 mV at 10 mA cm$^{-2}$ and a lower Tafel slope of 36.9 mV per decade. More importantly, without complex nanostructuring, the catalytic system is endowed with a good working stability. Our work offers a new insight for improving HER performance in the fields of photocatalysis, electrocatalysis, and photoelectrochemical catalysis. In view of the great variety of candidate materials, novel strategies could be foreseen to tune the catalytic activities.

## Methods

**Material synthesis**. First, 0.25 g of ferrous chloride (1.97 mmol) powder and 0.2 g of thiourea (2.63 mmol) were dissolved in 40 mL of ethylene glycol under vigorous stirring to form a homogeneous solution at 318 K. Then, 2 mL of sodium citrate aqueous solution (0.1 wt%) was dropped gradually in order to obtain good ultrathin FeS nanosheets in the following solvothermal process, and the mixture was stirred under the argon current for 30 min at room temperature to ensure complete reaction (step 1). Meanwhile, the woven CFC was washed sequentially with $H_2SO_4$ (5 M), acetone, deionized water, and ethanol under sonication for 0.5 h each to thoroughly remove organic residues and other impurities. Subsequently, the FeS nanosheets were grown on the cleaned CFC substrate by a solvothermal synthetic route. In brief, the pre-cleaned CFC substrates were vertically aligned in the above mixed solution under ultrasonication for 5 min. Thereafter, the homogeneous solution with CFC was transferred to a 50-mL Teflon-lined stainless-steel autoclave and heated at 483 K for 15 h. After naturally cooling down to room temperature, the CFC loaded with FeS nanosheets was taken out, consecutively washed several times with ethanol and deionized water, respectively, and then dried in a vacuum oven at 323 K for 8 h (step 2). The annealed process was employed to improve the purity and crystallization of the FeS nanosheets. Typically, the dry CFC loaded with FeS nanosheets was placed in a ceramic boat, which was inserted into the middle of a horizontal quartz tubular reactor. Before heating, the system was purged with 700 sccm (standard cubic centimeters per minute) high-purity argon (Ar 99.999%) for 1 h. After that, the furnace was heated at a rate of 2 K min$^{-1}$ to the temperature of 573 K and kept at this temperature for 2 h with a 5% $H_2S$/Ar (toxic by inhalation) flow of 200 sccm. The pressure was reduced to $1 \times 10^{-2}$ Torr for the duration of the reaction, and the lead nitrate solution was used to neutralize exhaust gas (step 3). The preparation route is schematically displayed in Supplementary Fig. 1.

**Characterization**. The morphology and microstructure were characterized by field-emission scanning electron microscopy (FE-SEM, Hitachi, S4800), high-resolution transmission electron microscopy (HR-TEM, JEOL-2100) equipped with X-ray energy-dispersive spectrum (EDS), tapping-mode AFM (Nanoscope IV4–1), XRD (XRD-7000, Shimadzu) with Cu K$_\alpha$ radiation ($\lambda = 0.15406$ nm), and X-ray photoelectron spectrometry (XPS, VG ESCALAB MKII). Fluke TiS45 infrared camera was used to take infrared photographs. The UV-vis absorption spectra were obtained by the diffuse reflection method on a spectrophotometer

(Varian Cary 5000) in the range from 250 to 2500 nm equipped with an integrated sphere attachment. The optoelectronic measurements were performed on the Keithley 4200 semiconductor characterization system in a Cascade Summit 12000 probe station. To acquire the contact angle measurements, a commercial contact angle system (DataPhysics, OCA 20) was used with 3 μL of water droplet as the indicator. In situ Raman spectra were recorded on a Raman microscope (NR-1800, JASCO) using a 514.5-nm argon ion laser with a temperature control table. Four-probe measurements of the longitudinal resistivity and Hall resistivity were conducted in a commercial cryostat (Quantum Design PPMS-14T). Differential scanning calorimetry (DSC) measurements were carried out on a PerkinElmer Diamond DSC under nitrogen atmosphere in aluminum crucibles with a heating or cooling rate of 10 K/min.

**EC measurement.** The electrochemical (EC) measurements were carried out on an electrochemical workstation (CHI 660D) in the standard three-electrode configuration with a graphite rod (Alfa Aesar, 99.9995%) as the counter electrode, commercial Ag/AgCl (saturated KCl) as the reference electrode, and CFC loaded with FeS nanosheets as the working electrode. Before the electrochemical measurement, the electrolyte (0.1 M KOH) was degassed by bubbling pure hydrogen for at least 20 min to ensure the $H_2O/H_2$ equilibrium. Linear-sweep voltammetry (LSV, $J$–$V$) was performed at a potential scanning rate of 5 mV s$^{-1}$, and electrochemical impedance spectroscopy (EIS) measurements were performed in the overpotential of 100 mV with a sinusoidal voltage of 5-mV amplitude in the range from 10 kHz to 0.1 Hz. The temperature control table equipped with a flow of cooling water was used to keep the temperature constant, and the temperature was recorded by a K-type thermocouple (resolution, 0.1 K). To further evaluate the HER kinetics, Tafel plots were obtained by re-plotting the polarization curves as overpotential ($\eta$) vs. log current (log $j$). The Tafel slopes were estimated by fitting the linear portion of the Tafel plots to the Tafel equation ($\eta = b \log(j) + a$). All the data were presented relative to the reversible hydrogen electrode (RHE) with iR compensation through RHE calibration described as follows. The external potentials [E(Ag/AgCl)] were measured against the Ag/AgCl reference electrode and converted to the potentials toward reversible hydrogen electrode RHE (E(RHE)) using the Nernst function:

$$E(RHE) = E(Ag/AgCl) + E°(Ag/AgCl) + 0.059\,pH \qquad (2)$$

where E°(Ag/AgCl) is the standard electrode potential of Ag/AgCl reference electrode (0.1976 V and 0.1735 V vs. RHE at 298 K and 323 K, respectively). The EC performance with the assistance of NIR light was also determined on an electrochemical workstation. The working electrode was irradiated at the front side with NIR light emitted from a Xe lamp with a 750-nm long-wave-pass filter. The NIR light density was uniformly calibrated to 40 mW cm$^{-2}$ on the working electrode. The linear-sweep voltammograms ($J$–$V$) were acquired at a scanning rate of 5 mV s$^{-1}$ under NIR light illumination, and the amperometric ($J$–$t$) curves were obtained under chopped light illumination at a bias voltage of −0.15 V vs. RHE to determine the EC properties. In the investigation of the EC stability, the photocurrent retention performance for 100 h at a bias voltage of −0.2 V vs. RHE and the amount of evolved $H_2$ was determined with an online gas chromatography (9790 II, Fuli, Zhejiang) equipped with a TCD detector and Ar gas carrier.

**Theoretical calculation.** The calculations were performed by using plane-wave basis Vienna ab inito simulation package (VASP) code in combination with the generalized gradient approximation (GGA)[37]. The GGA and Perdew–Burke–Ernzerhof (PBE) exchange-correlation functional with U = 1.0 eV is adopted to treat the localized $d$ orbits of the Fe atom[38–40]. We used projected augmented-wave (PAW) pseudopotentials, and in order to achieve a satisfactory degree of convergence, the plane-wave expansion has been truncated at a cutoff energy of 550 eV and the integrations over the Brillouin zone were performed considering $16 \times 16 \times 10$ and $10 \times 10 \times 6$ uniform Monkhorst–Pack k-point grids for the high-symmetry $P6_3/mmc$ and low-symmetry $P\bar{6}2_c$ unit cells, respectively. The (100) surfaces with Fe and S termination of $P6_3/mmc$ phase and its corresponding surface in $P\bar{6}2_c$ phase are both modeled as a $(1 \times 2)$ supercell consisting of a six-trilayer slab and a vacuum space of 20 Å to avoid the interaction between periodical images, and the bottom two trilayers are fixed to mimic the bulk. The Monkhorst–Pack k-point grid of $8 \times 8 \times 1$ is adopted, and the relaxation is carried out until all forces on the free ions are converged to 0.03 eV/Å. The activation energies were calculated using the climbing nudged elastic band (NEB) method[35]. Electromagnetic simulations were obtained by a finite element method (FEM).

## Data availability
The data that support the findings of this study are available from the corresponding authors on request.

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

## Acknowledgements

This work was supported by National Basic Research Programs of China under Grant Nos. 2018YFA0306004, 2017YFA0303200, National Natural Science Foundation (Nos. 21872071, 11674163, 11874204, 61574080, 61505085, and 11404162), and Natural Science Foundation of Jiangsu Province (BK20171332 and BK20161117). This work was also supported by the Fundamental Research Funds for the Central Universities (0204–14380066 and 0204–14380083) and High Performance Computing Centers of Nanjing University and Shenzhen. Partial support was from the Postgraduate Research and Practice Innovation Program of Jiangsu Province (KYCX17_0036).

## Author contributions

G.Z. performed the experiments and co-wrote this paper. Y.S., L.L.W. and Y.Y.H. prepared the samples and proofread the paper. J.H.G. and Y.G. plotted the figures. J.T.C. prepared the samples. J.C.S. and F.R.H. designed the experimental setup. L.Z.L. and X.L.W. analyzed the data, theoretical calculation, and wrote the paper.

## Additional information

**Competing interests:** The authors declare no competing interests.

**Journal Peer Review Information**: *Nature Communications* thanks Marion Giraud, and other anonymous reviewer(s) for their contribution to the peer review of this work. Peer reviewer reports are available.

