## [Peer Review File · Nature Communications]

Editorial Note: Parts of this peer review file have been redacted as indicated to maintain the confidentiality of unpublished data. When text is deleted in rebuttals and referee reports, add "[Redacted]" in that location.

Reviewers' comments:

Reviewer #1 (Remarks to the Author):

The manuscript described a way to improve HER performance via semiconductor-metal transition in ultrathin troilite FeS nanosheets vertically grown on carbon fiber cloth (CFC) triggered by near infrared (NIR) radiation at near room temperature. It can be accepted for publication in Nature Communication after a minor modification.

The key point to improve HER performance in this manuscript is to realize the phase transition in ultrathin troilite FeS nanosheets grown on CFC triggered by near infrared radiation. As we known, FeS has many crystal structures, such as Mackinawite, Cubic FeS, Troilite and pyrrhotite etc. The phase transition will change electron mobility and the carrier concentration will increase or decrease, but we know that the carrier concentration will increase under light radiation even if there is no phase transition. How to prove light radiation lead to the phase transition? Of course, in the manuscript, the authors used in situ Raman spectra to confirms that the E_g modes are sharply down-shifted due to phase transition at 403 K without NIR light radiation and 323 K with NIR light radiation. The authors should provide more evidence to verify the phase transition caused by light radiation. At least, the authors should measure the carrier concentrations before and after the phase transition in detail.

As a comparison, the authors should use electrochemical impedance spectra to check the change of charge transfer resistance of phase transition without NIR light radiation.

Reviewer #2 (Remarks to the Author):

Review on paper NCOMMS-18-27543 by Dr Liu and co-workers entitled "Photoinduced Reversible Semiconductor-Metal Transition in Ultrathin Troilite FeS Nanosheets to Trigger Efficient Hydrogen Evolution"

Summary

The manuscript addresses the synthesis and characterizations of a material consisting of FeS troilite nanosheets vertically grown on Carbon Fiber Cloth (CFC) and its use as a HER electrocatalyst in alkaline conditions. Its synthesis is performed in 3 steps using a polyol solvothermal solution synthesis followed by thermal treatment in H₂S. This FeS catalyst undergoes a semi-conductor (SC) to metal phase transition which thermally occurs around 400 K but this transition temperature can be shifted to ca. 323 K under NIR irradiation. The light-induced metallic FeS phase demonstrates an overpotential of 142 mV and a Tafel slope of 36.9 mV/decade and outperforms its semi-conductor FeS counterpart, from which it is generated. With the help of both experiments and theoretical calculations, the authors explains this activity enhancement by many factors, including specific morphology as well as improved electronic conduction of the catalyst.

The authors emphasizes on the following results: (i) Room temperature NIR-triggered reversible phase transition in FeS troilite nanosheets, (ii) High catalytic HER activity in KOH 0.1 M.

Comments on the manuscript:

For over a decade now, sulfides materials based on earth-abundant transition metals have been considered as promising alternatives to platinum as HER catalysts. As a consequence, the simple strategy presented in this article to further improve HER performances of such materials is interesting (SC to metal transition in a FeS troilite sample). Secondly, there is an abundance of characterizations

of the material and theoretical calculations. They are usually well conducted and support the claims but suffer from imprecision. Nevertheless, the intrinsic performances of the catalysts are not so impressive in terms of activity and stability and some important characterizations are missing. Many questions arise also mainly because the manuscript is often very imprecise and these issues should be addressed by the authors.

As a conclusion, this work deserves much more precisions and requires major revisions. In its current state, it seems not to be suited to such a high standard journal as Nature Communications. I will explain myself below.

Concerning typos and rewriting:

The paper contains a few typos listed below

- 1) Reference 3 is not correct: it should be *Angew. Chem. Int. Ed.*, 56, 7610-7614 (2017)
- 2) In the SI, page S2 previous studies should be corrected.
- 3) In the SI, equation 2 page S3, a "+" sign between ΔE_{H^+} and ΔE_{ZPE} seems missing.
- 4) In the synthesis description, 2mL should be separated.
- 5) In the synthesis description, hydrothermal should be replaced by solvothermal since the mixture consists of 40 mL of EG and 2 mL of water.

Concerning references.

A review dealing with sulfides materials based on earth-abundant transition metals for HER is missing such as *ACS Catal.* 2016, 6, 8069–8097 by S. Anantharaj et al. or any other helping to compare the performances of the described catalyst with state of the art metal sulfides.

Concerning adding more information:

- 1) The chemical suppliers must be mentioned in the experimental part for all the compounds used (reactants, CFC, KOH, gases, etc).
- 2) The number of moles should be added in the synthesis for FeCl₂ and thiourea.
- 3) Toxic compounds (thiourea, H₂S for instance) should be manipulated with care. This should be added in the manuscript.
- 4) A table containing all the performances of the studied materials vs HER (for instance onset potential and potential @10 mA/cm² and Tafel plots) and also some other iron-based sulfides in the literature should be added in SI for sake of clarity of the comparison.

Concerning experiments and discussion:

1) The value of 142 mV for the overpotential is never related to a current density in the manuscript. It seems logically to be the overpotential at 10 mA/cm² but it is never written. If so, this value is among fairly low values (according to the review of S. Anantharaj cited above) but does not compare favorably with the value of 58 mV obtained in reference 15 or 96 mV obtained in reference 7, or 105 mV obtained in the following article by Long, X. et al., *J. Am. Chem. Soc.*, 2015, 137, 11900 – 11903. The authors should be much more convincing about why the performances of their catalyst are so impressive, even if it is true the experiments are conducted in alkaline solution and not acidic.

2) In the title, abstract and main conclusion, the authors emphasize on the reversibility of this transition (evidenced on the DSC curves on fig. S4b)? Is it an advantage or a drawback since the metallic phase cannot be stabilized in the dark?

3) How the critical NIR-triggered transition temperature of 323 K was determined? At 298 K the NIR-induced transition does not occur but what happens between these two values?

4) According to the authors, this synthesis is original because neither published elsewhere nor adapted

from a publication. So how to account for the role of the very small amount of citrate in the synthesis? Why should it be added dropwise and which reaction completes at this step (it should be mentioned)?

5) In the synthesis, how crystallinity and purity of FeS is improved by the thermal treatment? XRD of the sample before and after H₂S exposure should be presented. Is the FeS non stoichiometric before the treatment (usually FeS is Fe-deficient)?

6) The label "vertically aligned" for the nanosheets is misleading especially to interpret figure S10. I guess it means that the long distance of each sheet is perpendicular to the main axis of the carbon fiber but it not straightforward and should be clarified. Moreover on S10 fig., the middle drawings (top and bottom) are not very useful and directions P and S must appear for sake of clarity.

7) The experiments have been conducted in KOH only. Why not in acid medium since performances are expected to be even better?

8) Electrochemical surface area (ECSA) experiments should be performed to compare real electroactive area and accessible active sites of the SC FeS troilite phase with the metallic one, especially since this phase can only be accessible under irradiation and since a structural transition occurs. This may have a strong impact on ESCA.

9) XPS experiments should be very instructive to better describe the surface state of Fe and Sulfur. These experiments should be performed before and after catalysis. Indeed, polyol synthesis of metal sulfides is known for instance to give surface sulfate moieties which could impact catalysis.

10) On figure 4h, there is no mention of any irradiation wavelength. It seems the test was done under NIR irradiation but it is unclear. The wavelength must be mentioned and the power as well. This 10 h stability test (fig 4h) is rather convincing but should be extended to a longer period (80 h for example in reference 15). The post-catalysis characterization should not only be limited to a SEM analysis (fig. S17). The crystallographic structure and chemical composition (XRD and EDS) must be also investigated.

11) The Tafel slope of the metallic phase was determined to be 36.9 mV/decade which "suggests that Tafel-Volmer mechanism plays a predominant role in determining the HER rate and the recombination step is the rate limiting step", as the authors said. This is what is usually admitted. However, on fig. 5, the result of VASP calculations seems to be in contradiction with this sentence. If I understand well this reaction scheme, the activation barrier of the desorption step is much lower than that of each adsorption state of H₂O and formation of surface H. Is there any explanation to account for this difference?

12) What is the amount of FeS deposited on the CFC? Could the authors provide a "intrinsic" activity of the catalysts, that is the current density per mg of catalyst at a given over potential (for instance $\eta = -0.1V$)?

13) The control figure S9 is not convincing and should be improved by adding the curves corresponding to the same experiments but performed at 328.4 K to evidence the absence of change.

14) Values of the fits for impedance spectroscopy (fig. 4g) should be presented in a table in SI, especially the changes in RCT values.

Reviewer #3 (Remarks to the Author):

The authors have realised the semiconductor-metal transition of troilite FeS by using near infrared radiation. Remarkably, the resulting metallic phase exhibits higher catalytic activity than its semiconductor counterpart.

Below my comments and suggestions:

To ensure reproducibility the authors should include the following computational details:

- The value U of the Hubbard parameter
- Which among the three terminations (J. Phys. Chem. C 122, 24, 12810-12818) of the (100) prismatic surface of troilite has been used in the calculations
- The thickness of the surface slab in terms of FeS layers
- The bulk lattice vectors used to construct the surface unit cells

Please note that the convergence of the plane wave cutoff and k-point grid is not related to the convergence of the forces on the ions, which seems to be within 0.03 eV/Å. Thus, the expression "in order to achieve a satisfactory degree of convergence (0.03 eV/Å)" should be avoided and the ion forces threshold of 0.03 eV/Å simply listed.

FeS nanoparticles have already been reported to exhibit molecular hydrogen evolution in neutral water at room temperature (ACS Catal. 4, 2, 681-687). The authors should acknowledge the study above and try to compare the values of overpotential, exchange current density and Tafel slope reported there to those of the room temperature semiconductor FeS phase of this work.

The authors have indicated that the phase at high temperature is hexagonal (P63/mmc). However, while most of the works point towards that direction (J. Phys. Chem. Solids 2017, 111, 317-323 or Science 1995, 268, 1892), the appearance of a phase with an orthorhombic cell (Pnma) has also been suggested (J. Solid State Chem. 1990, 84, 194-210 or Am. Mineral. 1992, 77, 391-398). Do the authors have data supporting the P63/mmc phase? Can they comment on this point?

The title indicates a reversibility of the transition. Can the authors discuss it somewhere also in the manuscript?

Line 56: "Herein, for the first time, we realize the photoinduced reversible semiconductor-metal transition at near room temperature in ultrathin troilite FeS nanosheets vertically grown on carbon fiber cloth." At this point of the manuscript, the general reader may not know what exactly should be considered as novel in this work ("for the first time"). The reversibility? The transition? The room temperature? The nanosheets? The carbon fiber cloth substrate? A combination of them? Can the authors clarify? The abstract should be changed as well.

Line 87: "which is in consistence with the theoretical simulation" should be replaced by "which is in consistence with the structure of troilite"

Line 180: "From the difference in charge density of the two phases in upper panel in Fig. 4b, we find the change only happens in charge density localized on Fe atoms (red region), showing Fe atoms has higher catalytic activity." Can the authors clarify this sentence? It is not clear to me why the change in charge density should give a higher catalytic activity.

Line 183: "This agrees with Fig. 4a because more electrons are transferred into these Fe sites to optimize the adsorption of hydrogen (Fig. S12)." What I see in Figure 4a is that with the phase transition the Gibbs free energy of the adsorbed H on Fe1 (Fe2) increases (decreases) to zero (which is beneficial), but I do not understand how this correlates with Fig. S12, which shows that both Fe1 and Fe2 gain charge. In addition, I am a bit surprised by the fact that Fe1 and Fe2 give opposite signs for the adsorption energy of H. Can the authors comment on this? Please note that it is not easy to visualise the different arrangement of Fe1 and Fe2 from the upper panel of Fig. 4b. Finally, I do not believe that the results on the charge transfer, i.e. the upper panel of Fig. 4b and Fig. S12 strengthen the manuscript and therefore they may be removed, while, a better figure with the Fe1, Fe2 and S the labels should be added.

Line 184: "The atomic density of state (DOS) analysis confirms that the highest occupied orbital is composed by not p-states of S atoms but d-states of Fe atoms (lower panel in Fig. 4b), which is benefit for optimizing the electronic structures to improve HER performance (see detailed orbital resolved DOSs in Figs. S13 and S14)." Can the authors clarify this argument?

Line 202: "Moreover, such low Tafel slope suggests that Volmer Tafel mechanism plays a predominant role in determining the HER rate and the recombination step is the rate-limiting step". I believe that the Tafel slopes of the curves of Fig. 4d may suggest instead a Volmer-Heyrovsky mechanism (J. Am. Chem. Soc. 2015, 137, 7365-7370 or J. Mater. Chem. A, 2015, 3, 1494). Why has it been excluded?

Line 235: "The energy along semiconductor phase (P62c) reaction pathway (marked by red line) is larger than that along metal phase (P63/mmc) (marked by black line), indicating that metal phase has obvious advantages in HER process." This sentence has little meaning in my opinion. From Fig. 5, it is hard to infer why the efficiency of the metal phase should be higher than that of the semiconductor, as the activation barriers look very similar. I suggest the authors to remove this sentence from the manuscript and to use Fig. 5 only to speculate a possible Volmer-Tafel pathway for the two phases. However, the authors should compare it with that of an alternative Volmer Heyrovsky mechanism. Please see my comment above.

Line 250: "Therefore, the semiconductor-metal phase transition plays a critical role in improving HER performance". I do not think that this conclusion is supported by Fig. 5, and therefore it should be removed from the text.

"Photoinduced Reversible Semiconductor-Metal Transition in Ultrathin Troilite FeS Nanosheets to Trigger Efficient Hydrogen Evolution"

ID: NCOMMS-18-27543

Response to the report of the reviewer 1

Comment 1. *The manuscript described a way to improve HER performance via semiconductor-metal transition in ultrathin troilite FeS nanosheets vertically grown on carbon fiber cloth (CFC) triggered by near infrared (NIR) radiation at near room temperature. It can be accepted for publication in Nature Communication after a minor modification.*

Question: The key point to improve HER performance in this manuscript is to realize the phase transition in ultrathin troilite FeS nanosheets grown on CFC triggered by near infrared radiation. As we known, FeS has many crystal structures, such as Mackinawite, Cubic FeS, Troilite and pyrrhotite etc. The phase transition will change electron mobility and the carrier concentration will increase or decrease, but we know that the carrier concentration will increase under light radiation even if there is no phase transition. How to prove light radiation lead to the phase transition? Of course, in the manuscript, the authors used in situ Raman spectra to confirm that the E_g modes are sharply down-shifted due to phase transition at 403 K without NIR light radiation and 323 K with NIR light radiation. The authors should provide more evidence to verify the phase transition caused by light radiation. At least, the authors

should measure the carrier concentrations before and after the phase transition in detail.

Answer: Thanks for the reviewer's comments and suggestions. To further verify the phase transition caused by light radiation, the electronic conductivity versus temperature with and without NIR light radiation were carried out and shown in Figure R1-1. Generally, the electronic conductivity of metal phase is higher than semiconductor phase. Different from the linear change in dark, the values of electronic conductivity sharply increase around 330 K, as shown in Figure R1-1a, which only can be ascribed to appearance of semiconductor-metal transition. This is because the electronic conductivity is only slightly enhanced due to the NIR light radiation, similar to the changes in 290-310 K. The calculated electronic density of state (DOS) in Figure R1-1b demonstrates that the $P6_3/mmc$ phase has obvious metal feature but $P\bar{6}2_c$ phase displays a semiconductor property. By comparing electron localization functions of two phases in Figures R1-1 (c) and (d), we also found that the electrons have smaller tendency to locate around pristine atoms in metallic $P6_3/mmc$ phase (Figure R1-1d) than those in semiconductor $P\bar{6}2_c$ phase (Figure R1-1c), which can be used to explain soaring electronic conductivity via phase transition. Therefore, the mutation of electronic conductivity induced by NIR light radiation can be regarded as another evidence to support our proposed phase transition.

[Redacted]

In addition, the photocurrent variation induced by NIR light radiation were carried out to evaluate the photogenerated carrier concentrations before and after phase transition in Figure R1-2a and R1-2b, respectively. The measured results indicate that the photocurrent after phase transition in Figure R1-2b is obviously higher than that before phase transition in Figure R1-2a. According to the formula (1) and the carrier mobility (Fig. S10 and Fig. S11), we found that the photogenerated carrier concentration changes from $N=7.79 \times 10^{12}/\text{cm}^3$ (before phase transition) to $N=9.28 \times 10^{12}/\text{cm}^3$ (after phase transition), leading to about ten times increment. The theoretical calculation in Figure R1-2c discloses that the phase transition potential barrier height will be reduced to zero if the photogenerated carrier concentration is enhanced by 5.8 times. Therefore, the dramatic increase in photogenerated carrier concentration (ten times) allows the realization of semiconductor-metal phase transition.

Figure R1-2. The I - V curves of photodetector without and with NIR irradiation before (a) and after (b) phase transition. (c) The potential barrier height versus the multiple of carrier concentration.

In order to more intuitively display the phase transition process, the potential barriers versus temperature with and without NIR light irradiation are schematically displayed in Figure R1-3. The transition potential barrier from $P\bar{6}2_c$ to $P6_3/mmc$ phase is obviously reduced by accumulating photogenerated carriers at the same temperature, which finally make the phase transition temperature decrease from 393 K to 323 K.

Figure R1-3. Schematic representation for potential barrier versus temperature with and without NIR light irradiation.

Comment 2. *As a comparison, the authors should use electrochemical impedance spectra to check the change of charge transfer resistance of phase transition without NIR light radiation.*

Answer: Thanks for the reviewer's comments and suggestion. We have fitted impedance spectroscopy to evaluate the values of R_s and R_{ct} with and without NIR light radiation, and the acquired values are presented in Table R1-1. The R_{ct} value at 298 K is slightly reduced from 41.51 to 35.94 Ω due to NIR light irradiation, but the value at 323 K decreases nearly by half. Associated with substantially theoretical and experimental results in this manuscript, we can conclude that the photoinduced semiconductor-metal phase transition is responsible for significant decrease of R_{ct} value at 323 K.

Table R1-1. Results of fitting Nyquist plots by equivalent circuit.

Sample	R_s (Ω)	R_{ct} (Ω)
323 K Light	1.15	13.12
323 K Dark	1.17	30.62
298 K Light	1.18	35.94
298 K Dark	1.21	41.51

These revised parts have been inserted into modified manuscript, such as page 6 lines 122-124, page 7 lines 139-144, and pages S25-S26, S16, S34.

End

Response to the report of the reviewer 2

Comment 1. *Summary*

The manuscript addresses the synthesis and characterizations of a material consisting of FeS troilite nanosheets vertically grown on Carbon Fiber Cloth (CFC) and its use as a HER electrocatalyst in alkaline conditions. Its synthesis is performed in 3 steps using a polyol solvothermal solution synthesis followed by thermal treatment in H₂S. This FeS catalyst undergoes a semi-conductor (SC) to metal phase transition which thermally occurs around 400 K but this transition temperature can be shifted to ca. 323 K under NIR irradiation. The light-induced metallic FeS phase demonstrates an overpotential of 142 mV and a Tafel slope of 36.9 mV/decade and outperforms its semi-conductor FeS counterpart, from which it is generated. With the help of both experiments and theoretical calculations, the authors explain this activity enhancement by many factors, including specific morphology as well as improved electronic conduction of the catalyst. The authors emphasize on the following results: (i) Room temperature NIR-triggered reversible phase transition in FeS troilite nanosheets, (ii) High catalytic HER activity in KOH 0.1 M.

Comments on the manuscript:

For over a decade now, sulfides materials based on earth-abundant transition metals have been considered as promising alternatives to platinum as HER catalysts. As a consequence, the simple strategy presented in this article to further improve HER performances of such materials is interesting (SC to metal transition in a FeS troilite sample). Secondly, there is an abundance of characterizations of the material and theoretical calculations. They are usually well conducted and support the claims but

suffer from imprecision. Nevertheless, the intrinsic performances of the catalysts are not so impressive in terms of activity and stability and some important characterizations are missing. Many questions arise also mainly because the manuscript is often very imprecise and these issues should be addressed by the authors.

As a conclusion, this work deserves much more precisions and requires major revisions. In its current state, it seems not to be suited to such a high standard journal as Nature Communications. I will explain myself below.

Answer: We highly appreciate the appropriate comments of the reviewer on our work and we have made major revisions to improve the quality of the manuscript. In addition, some typographical errors and confusing sentences have been revised. These revised parts are highlighted by red.

Comment 2. *Concerning typos and rewriting:*

The paper contains a few typos listed below

1) *Reference 3 is not correct: it should be Angew. Chem. Int. Ed., 56, 7610-7614 (2017)*

Answer: Thanks for the reviewer's help. This reference has been corrected.

Comment 3. *2) In the SI, page S2 previous studies should be corrected.*

Answer: The space between "previous" and "studies" has been added.

Comment 4. *3) In the SI, equation 2 page S3, a "+" sign between ΔE_{H^+} and ΔE_{ZPE}*

seems missing.

Answer: The sign of “+” has been revised in the revised manuscript.

Comment 5. *4) In the synthesis description, 2mL should be separated.*

Answer: The correct form for 2 mL has been used in the revised manuscript.

Comment 6. *5) In the synthesis description, hydrothermal should be replaced by solvothermal since the mixture consists of 40 mL of EG and 2 mL of water.*

Answer: The “hydrothermal” has been replaced by “solvothermal” in the revised manuscript.

Comment 7. *Concerning references.*

A review dealing with sulfides materials based on earth-abundant transition metals for HER is missing such as ACS Catal. 2016, 6, 8069–8097 by S. Anantharaj et al. or any other helping to compare the performances of the described catalyst with state of the art metal sulfides.

Answer: The references about HER performance, such as *ACS Catal. 2016, 6, 8069–8097*, have been added into our manuscript as Ref. 15, and the other references concerned have been added into SI and manuscript, and marked by red.

Comment 8. *Concerning adding more information:*

1) The chemical suppliers must be mentioned in the experimental part for all the compounds used (reactants, CFC, KOH, gases, etc).

Answer: All the chemical suppliers for chemical reactants, experimental materials and gases have been mentioned in the supporting information (see the part of chemicals and reagents, in page S4, marked by red).

Comment 9. 2) *The number of moles should be added in the synthesis for FeCl₂ and thiourea.*

Answer: The number of moles for FeCl₂ (1.97 mmol) and thiourea (2.63 mmol) has been provided in the supporting information (see the part of synthesis of FeS nanosheets, in page S5, marked by red).

Comment 10. 3) *Toxic compounds (thiourea, H₂S for instance) should be manipulated with care. This should be added in the manuscript.*

Answer: Thanks for the reviewer's kind suggestion. We have added the hazards statement (Toxic by inhalation, in contact with skin and if swallowed) for the thiourea, acetone and H₂S in the method section (see the part of synthesis of FeS nanosheets, in page S6 line 112, marked by red).

Comment 11. 4) *A table containing all the performances of the studied materials vs HER (for instance onset potential and potential @10 mA/cm² and Tafel plots) and also some other iron-based sulfides in the literature should be added in SI for sake of clarity of the comparison.*

Answer: Thanks for the reviewer's kind suggestion. The HER performances of FeS/CFC catalysts with and without NIR irradiation were collected and shown in

Table R2-1. The results indicate that the onset potential, the over potential and the Tafel slope are all reduced by NIR light irradiation, especially at 323 K.

Table R2-1, HER parameters on FeS/CFC catalyst under different conditions.

Sample	Onset potential (mV vs. RHE) ^a	Over potential (mV vs. RHE) ^b	Tafel slope (mV per decade)
323 K Light	94	142	36.9
323 K Dark	101	199	51.8
298 K Light	102	192	51.4
298 K Dark	116	222	61.1

a. Onset potential is for achieving 0.5 mA cm^{-2} ; **b.** over potential is for achieving 10 mA cm^{-2} .

These parts have been added into page 10 line 205, and renamed as Table S1 in page S32.

In addition, the HER performances of previously reported Fe-based electrocatalysts were collected in Table R2-2 for comparison. The results indicate that our synthesized FeS/CFC catalysts have the lowest Tafel slope and excellent stability.

Table R2-2. The comparison of HER performances between FeS/CFC and previously reported Fe-based electrocatalysts.

Catalyst	Electrolyte solution	Overpotential (mV vs. RHE)	Exchange current density (j_0 , $\mu\text{A cm}^{-2}$)	Tafel slope (mV per decade)	Stability test	Reference
FeS/CFC	0.1 M KOH	142	3.1	36.9	100 h	This work
FeS ₂ /CoS ₂	1.0 M KOH	78.2	–	44	80 h	Small, 2018, 14, 1801070
A-FeNiS	0.5 M H ₂ SO ₄	105	2.2	40	40 h	J. Am. Chem. Soc. 2015, 137, 11900–11903
NiFe/NiCo ₂ O ₄ /NF	1.0 M KOH	105	470	88	10 h	Adv. Funct. Mater. 2016, 26, 3515
Ni-Fe/NC	1.0 M KOH	219	–	110	1200 s	ACS Catal. 2016, 6, 580
Meso-FeS ₂	0.1 M KOH	96	630	78	24 h	J. Am. Chem. Soc. 2017, 139, 13604
pyrite FeS ₂ film	0.5 M H ₂ SO ₄	270	–	62.5	–	Energy Environ. Sci., 2013, 6, 3553
Pyrite FeS ₂ /C	0.1 M phosphate buffer	557	–	204	–	ACS Catal. 2016, 6, 2626
Ni Fe LDH-NS DG10	1.0 M KOH	300	–	110	20000 s	Adv. Mater. 2017, 29, 1700017
FeS ₂ @MoS ₂ /rGO	0.5M H ₂ SO ₄	123	17.5	38.4	8 h	Chem. Commun., 2016, 52, 11795
FeS P	0.5M H ₂ SO ₄	160	–	44	–	ACS Catal. 2017, 7, 4026
2-cycle NiFeO _x /CFP	1.0 M KOH	88	–	150	100 h	Nature Comm. 2015, 6, 7261
FeS	1.0 M Phosphate buffer	350	0.66	150	–	ACS Catal. 2014, 4, 681
FeCoNi-HNT	1.0 M KOH	58	–	37.5	100 h	Nature Comm. 2018, 9, 2452
(Fe _{0.48} Co _{0.52})S ₂	0.5 M H ₂ SO ₄	196	0.959	47.5	20 h	J. Phys. Chem. C 2014, 118, 21347.
Fe-CoP/Ti	1.0 M KOH	78	–	75	20 h	Adv. Mater. 2017, 29, 1602441
Fe-N-C	0.5 M H ₂ SO ₄	130	27	89	5 h	Adv. Energy Mater. 2018, 8, 1701345
Ni-Fe-P porous nanorods	1.0 M KOH	79	–	92.6	24 h	J. Mater. Chem. A, 2017, 5, 2496
Co _{0.6} Fe _{0.4} P/C NT	0.5 M H ₂ SO ₄	67	–	57	24 h	Adv. Funct. Mater. 2017, 27, 1606635
FeP NAS/CC	1.0 M KOH	218	–	146	20 h	ACS Catal. 2014, 4, 4065

These parts have been added into page 10 lines 216-218, and renamed as Table S2 in page S33.

Comment 12. *Concerning experiments and discussion:*

1) The value of 142 mV for the overpotential is never related to a current density in the manuscript. It seems logically to be the overpotential at 10 mA/cm² but it is never written. If so, this value is among fairly low values (according to the review of S. Anantharaj cited above) but does not compare favorably with the value of 58 mV obtained in reference 15 or 96 mV obtained in reference 7, or 105 mV obtained in the following article by Long, X. et al., J. Am. Chem. Soc., 2015,137,11900 –11903. The authors should must be much more convincing about why the performances of their catalyst are so impressive, even if it is true the experiments are conducted in alkaline solution and not acidic.

Answer: Thanks for the reviewer's kind suggestions. The overpotential of 142 mV is related to 10 mA/cm² in our work, in page 13, which has been explained in the revised manuscript. Reference 15 and previous report by Long, X. et al. (J. Am. Chem. Soc., 2015, 137, 11900 –11903) obtain the values of 58 mV and 105 mV, better than ours, because Co and Ni elements are introduced into their system to enhance the HER performance. It is generally accepted that the common existence of Fe, Co and Ni elements can produce better catalytic activity [ACS Catal. 2016, 6, 8069–8097]. If nickel or cobalt is intentionally doped into our system, the hydrogen evolution performance would be greatly improved (this work will be detailedly discussed in our next report). The main reason why the value of 96 mV was obtained in reference 7 is the difference in substrate. Compared with CFC substrate, Ni foam substrate has stronger catalytic activity. If CFC is replaced by a better substrate, e.g., iron foam in

previous work by Zou. et al., Chem, 4, 1139-1152, a better catalytic activity can be achieved.

In this work, we mainly focus on a simple and cheap light-induced phase transition strategy to improve intrinsic catalytic activity of FeS/CFC instead of traditional methods such as constructing various nanostructures, introducing vacancies or dopants and so on. Our findings also provide new inspirations for the steering of electron transfer and designing new-type catalysts. In order to better clarify this physical mechanism, the studying system cannot be designed so complex to exclusively chase high HER performance. Even so, the comprehensive comparison in Table R2-2 confirms that our FeS catalysts have the lowest Tafel slope, the excellent stability, and good catalytic activity. Thanks for the reviewer's kind suggestion again, to improve the HER performance of FeS by doping single atom Co and Ni has been carried out in our next work.

These parts have been added into page 10 lines 216-218, and Table S2 in page S33.

The reference concerned has been added as Ref. 34.

Comment 13. 2) *In the title, abstract and main conclusion, the authors emphasize on the reversibility of this transition (evidenced on the DSC curves on fig. S4b)? Is it an advantage or a drawback since the metallic phase cannot be stabilized in the dark?*

Answer: The reversibility of this transition is the material's intrinsic feature, which can provide some new inspirations for steering of electron transfer and designing new-type catalysts, such as optoelectronic devices for information storage. This

design can also be widely used to other catalysts to enhance its original HER performance by simple NIR light irradiation because the electron transfer efficiency may be enhanced by smaller potential barrier. In addition, this reversibility can make catalyst restore to initial state easily, and endow HER with good temporal and spatial control as well. Therefore, although the reversibility shows some misunderstanding about HER performance, it is still mentioned in the manuscript.

In this work, we focus on the improvement of HER performance by a simple photoinduced semiconductor-metal phase transition. In order to highlight the theme and consider the suggestions of both the Reviewer 2 and the Reviewer 3, the emphasized descriptions about “reversibility of this transition” in title, abstract and conclusion have been removed partially.

Comment 14. 3) *How the critical NIR-triggered transition temperature of 323 K was determined? At 298 K the NIR-induced transition does not occur but what happens between these two values?*

Answer: Thanks for the reviewer’s comments. In this work, the NIR-triggered transition temperature is determined by Raman mode shifts. The *in situ* Raman spectra in Figure 2d disclosed that the B_{1g} peak remains unchanged at 218 cm⁻¹, but E_g mode slightly down-shifts from 285 cm⁻¹ (283 K) to 281 cm⁻¹ (393 K), then rapidly shifts to 277 cm⁻¹ (403 K) and keeps near constant at ~413 K as temperature increases. When the NIR light radiation is introduced, the phase transition temperature of

ultrathin FeS nanosheets decreases to 323 K because E_g mode under NIR light radiation at 323 K down-shifts to the same position of that at 403 K without NIR light. To more clearly determine the transition temperature, the peak shifts of E_g modes at different temperatures observed from *in situ* Raman spectra are displayed in Figure R2-1. It is generally accepted that the shift of Raman mode corresponds to structural transition. Therefore, compared with that of without NIR light irradiation, the sharp shifts of E_g modes induced by NIR light irradiation at initial stage (~ 323 K, marked by red lines) make us determine the transition temperature facily.

Figure R2-1. Peak shifts of E_g modes at different temperatures observed from *in situ* Raman spectra.

The corresponding discussion has been inserted into page S14.

In order to more intuitively display the phase transition process, the potential barrier versus temperature with and without NIR light irradiation is schematically displayed in Figure R2-2. The transition potential barrier from $P\bar{6}2_c$ to $P6_3/mmc$ phase

is obviously reduced by accumulating photogenerated carriers at the same temperature, which finally makes the phase transition temperature decrease from 393 K to 323 K. When the temperature is lower than 323 K, accumulating photogenerated carriers can partially decrease the potential barrier height but cannot lead to phase transition.

Figure R2-2. Schematic representation for potential barrier versus temperature with and without NIR light irradiation.

This part has been added into page S16 and page 6 lines 123-124, and renamed as Fig. S8.

Comment 15. 4) According to the authors, this synthesis is original because neither published elsewhere nor adapted from a publication. So how to account for the role of the very small amount of citrate in the synthesis? Why should it be added drop wise and which reaction completes at this step (it should be mentioned)?

Answer: Thanks for the reviewer's kind suggestions. The citrate molecules can adsorb preferentially on the (001) plane of FeS crystal nucleus to modify its polarity. Thus the c-axis growth is drastically suppressed compared to the equatorial growth, leading to the better formation of ultrathin FeS sheet agglomerates. Besides, the slow

addition of citrate make it well-distributed to combine with Fe²⁺ ion, avoiding conglomeration of FeS in following solvothermal process, which has obvious advantages in obtaining our expected ultrathin FeS nanosheet.

The corresponding descriptions have been added into page S5.

Comment 16. 5) *In the synthesis, how crystallinity and purity of FeS is improved by the thermal treatment? XRD of the sample before and after H₂S exposure should be presented. Is the FeS non stoichiometric before the treatment (usually FeS is Fe-deficient)?*

Answer: Thanks for the reviewer's kind suggestions. The crystallinity and purity of FeS is improved by the thermal treatment. As seen in Figure R2-3, some weak XRD peaks originating from iron deficiencies or other iron-sulfur compounds will be removed by subsequent thermal treatment in H₂S/Ar atmosphere. EDS analysis confirms that the Fe/S ratio is enhanced to around 1:1 (after annealing) from 0.83:1 (before annealing) via the thermal treatment.

Figure R2-3. XRD patterns of FeS nanosheets before and after thermal

treatment in H₂S/Ar atmosphere.

These results have been added into page 4 line 78 and page S12, and renamed as Figure S4.

Comment 17. 6) *The label “vertically aligned” for the nanosheets is misleading especially to interpret figure S10. I guess it means that the long distance of each sheet is perpendicular to the main axis of the carbon fiber but it not straightforward and should be clarified. Moreover on S10 fig., the middle drawings (top and bottom) are not very useful and directions P and S must appear for sake of clarity.*

Answer: Thanks for reviewer’s suggestion. We have added some illustrations to eliminate confusions about Figure S10. Firstly, as illustrated in Figure S10, the drawings of P and S wave have been updated to clarify the simulation model briefly and straightforward. The models of FeS nanosheets perpendicular to the main axis of the carbon fiber (top) and accumulated (core–shell) structure (bottom) are displayed in Figure R2-4a-c and Figure R2-4d-f, respectively. Besides, the incident plane waves of different polarization orientations, such as 0°, 45° and 90°, are given to study the effects of polarization state of light. The middle drawings of top and bottom are just utilized to indicate a common polarization direction, which can be decomposed into P-waves and S-waves based on Jones vector theory.

Figure R2-4. Simulation of polarization absorption characteristics based on Finite-Difference Time-Domain (FDTD) method.

These revised parts have been added into page S22, and this figure is renamed as Figure S13.

Comment 18. 7) *The experiments have been conducted in KOH only. Why not in acid medium since performances are expected to be even better?*

Answer: Thanks for the reviewer's suggestion. The FeS can really exhibit better HER performance in acid medium as shown in Figure R2-5a, however, the stability in acid medium cannot compare with that in alkaline medium, as shown in Figure R2-5b. After 100 hours stability test, the HER performance reduces by 40.6% in acidic medium. Compared with that in acidic medium, the degradation efficiency of FeS/CFC remains almost unchanged in alkaline (91.9 %) and neutral medium (93.2%). The excellent stability and recyclability in alkaline solution is significant for practical applications.

As we all know, to design the recyclable catalysts in alkaline solutions is more compatible with current hydrogen production technology because the scalable and sustainable production of hydrogen fuel through efficient and cost-effective electrocatalytic water splitting technologies such as water-alkali and chlor-alkali electrolyzers, is highly promising as a means to meet the future global energy demands.

[Redacted]

Comment 19. 8) *Electrochemical surface area (ECSA) experiments should be performed to compare real electroactive area and accessible active sites of the SC FeS troilite phase with the metallic one, especially since this phase can only be accessible under irradiation and since a structural transition occurs. This may have a strong impact on ESCA.*

Answer: Thanks for the reviewer's suggestion. To assess the electrochemical active surface area (ECSA), double layer capacitance (C_{dl}) of the catalyst was measured by a simple cyclic voltammetry method (Figure R2-6a) to roughly calculate the value of ECSA. The current density at the selected potential from the regions of no Faradaic processes shows the linear correlation with the scan rate, and the slope of the fitting curve is considered as the C_{dl} . In Figure R2-6b, the C_{dl} of $P6_3/mmc$ phase indicates ECSA value of 552.5 cm^2 , which is about 4.5 times larger than that of $P\bar{6}_2c$ phase

(112.5 cm²), suggesting the enhanced catalytic active sites for HER. In consequence, the increased catalytic active sites result in the significant improvement of catalytic performance.

To measure electrochemical double-layer capacitance (C_{dl}), the potential was swept nine times at each scan rate (10, 20, 30, 40, 50, 60, 70, 80, and 90 mV/s) in the scan range from 0.10 to 0.20 V vs. RHE. Capacitive currents were measured in a potential range where no faradic processes were observed. The measured capacitive current difference (ΔJ) at 0.15 V vs. RHE was plotted against scan rate and specific capacitance was determined from the slope of the linear fitting. The C_{dl} values for $P\bar{6}2_c$ phase and $P6_3/mmc$ phase are calculated to be 4.9 and 22.1 mF·cm⁻², respectively. The specific capacitance is converted into an electrochemical surface area (ECSA) using the specific capacitance value for a flat standard with 1 cm² of real surface area. We use the specific capacitance (20–60 $\mu\text{F cm}^{-2}$) of 40 $\mu\text{F cm}^{-2}$ here to calculate the ECSA (Nat. Commun. 2018, 9, 2452), according to the following Eq:

$$\text{ECSA} = \frac{C_{dl}}{40 \mu\text{F}/\text{cm}^2} \text{cm}_{\text{ECSA}}^2$$

Figure R2-6 (a) FeS nanosheets under NIR light irradiation at 323K at various scan rates. (b) Charging current density difference plotted against scan rate at 0.15 V. The linear slope, equivalent to twice the double-layer capacitance, C_{dl} , was used to represent the ECSA.

These revised parts have been added into page 10 lines 213-216 and page S27-S28, and this figure is renamed as Figure S17.

Comment 20. 9) XPS experiments should be very instructive to better describe the surface state of Fe and Sulfur. These experiments should be performed before and after catalysis. Indeed, polyol synthesis of metal sulfides is known for instance to give surface sulfate moieties which could impact catalysis.

Answer: Thanks for the reviewer's suggestion. XPS spectra of Fe 2p and S 2p for FeS nanosheets before and after catalysis were acquired and shown in Figure R2-7. The spectra of Fe 2p for FeS nanosheets exhibit two peaks at 710.6 and 724.8 eV, which are assigned to the $2p_{3/2}$ and the $2p_{1/2}$ core levels of Fe^{2+} , respectively. Different sulfur species can be identified by the S 2p spectra in Figure R2-7(b). It is generally known that the major peaks at 162.1 and 163.2 eV could be ascribed to the disulfide ions (S^{2-}), and another peak at around 168 eV can be attributed to iron sulfate species (SO_4^{2-}) [J. Am. Chem. Soc. 2015, 137, 14023–14026]. The XPS spectra of S 2p before and after catalysis remain approximately unchanged, and no new XPS peaks corresponding to other sulfate moieties appear in our sample. In addition, comparison of X-ray photoelectron spectroscopy (XPS) results before and after catalysis for FeS

nanosheets illustrates that there are no discernible changes in the local crystal structure or chemical composition of FeS, which is further confirmed by SEM, XRD and EDS (see Comment 21). So the impact of surface sulfate moieties for catalysis can be negligible.

Figure R2-7. XPS spectra of (a) Fe 2p and (b) S 2p for FeS nanosheets before and after catalysis.

These revised parts have been added into page 11 lines 237-238 and page S30-S31, and this figure is renamed as Figure S19.

Comment 21. 10) On figure 4h, there is no mention of any irradiation wavelength. It seems the test was done under NIR irradiation but it is unclear. The wavelength must be mentioned and the power as well. This 10 h stability test (fig 4h) is rather convincing but should be extended to a longer period (80 h for example in reference 15). The post-catalysis characterization should not only be limited to a SEM analysis (fig. S17). The crystallographic structure and chemical composition (XRD and EDS)

must be also investigated.

Answer: Thanks for the reviewer's suggestion. The descriptions of irradiation wavelength (a Xe lamp with a 750 nm long-wave-pass filter) and power (40 mWcm^{-2}) have been added into EC experimental section in page S7. The stability measurement in Figure R2-8 indicates that the current density can commendably stabilize at 30 mA cm^{-2} for 100 h, which displays great potential for commercial utilization to replace noble metal materials. By comparison, there are no discernible changes in the local crystal structure or chemical composition of FeS, as confirmed by SEM, XRD and EDS in Figure R2-9.

Figure 2-8. Chronoamperometry curve (left) on the FeS/CFC at a constant potential of -0.2 V (versus RHE) accompanied by the continuous hydrogen measurement (right).

This longer stability testing result has been used to replace Fig. 4(h), and the corresponding description has been revised.

Figure 2-9 (a) FE-SEM image, XRD pattern (shown as the orange overlaid at the bottom portion of the panel), EDX spectrum (shown as the yellow overlaid at the top portion of the panel) and (b) EDS mapping of the FeS NSs/CFC electrode under long-term stability testing.

These revised parts have been added into page 11 lines 237-238 and page S30-S31, and this figure is renamed as Figure S19.

Comment 22. 11) *The Tafel slope of the metallic phase was determined to be 36.9 mV/decade which “suggests that Tafel-Volmer mechanism plays a predominant role in determining the HER rate and the recombination step is the rate limiting step”, as the authors said. This is what is usually admitted. However, on fig. 5, the result of VASP calculations seems to be in contradiction with this sentence. If I understand well this*

reaction scheme, the activation barrier of the desorption step is much lower than that of each adsorption state of H₂O and formation of surface H. Is there any explanation to account for this difference?

Answer: Thanks for the reviewer's kind comments, and we apologize for misleading to our conclusion due to inappropriate choices of potential energy zero position. In our previous calculation, the initial position of H₂O is considered too far away from FeS surface, which makes it difficult for H₂O to spontaneously adsorb onto Fe site at FeS surface. This is because that the Pauling electronegativity difference between H^{*} and Fe site is too small (only 0.23) to trigger stronger attraction in longer distance. In addition, we make use of Hubbard U parameter to correct exchange-correlation functional and norm-conserving pseudo-potentials of DFT calculation to obtain good constants in perfect agreement with experiment (0.1 %). However, this correction makes *d* orbital electrons strongly localized near Fe atoms by screen-exchange interaction, leading to weak interaction between Fe and other elements. For example, the lattice constant is enlarged to a=3.44 Å; b=5.85 Å for U=1 eV, from a=3.40 Å, b=5.79 Å for U=0 eV. Therefore, the initial distance between H₂O and FeS surface should be reduced to coincide with the physical truth, otherwise an extra spatial potential barriers will be inevitably counted into the Volmer process. Therefore, this amplified Volmer potential energy induced by calculated method cannot truly reflect our experimental conclusion and simultaneously cause some misunderstandings of reviewers. According to the Reviewer 3' suggestion and recent report [Nat. Commun., 2018, 9, 1425], we have improved our calculation method, such as reconsidering potential energy zero position, choosing a suitable adsorptive sites, to compare the

reaction process between the Volmer-Heyrovsky and the Volmer-Tafel mechanism, as shown in Figure R2-10. The recalculated results indicate that the rate limiting step is mainly determined by relative energy of Volmer-Heyrovsky process (electrochemical desorption step), which also can be used to eliminate the Reviewer 2’s confusion. As a general feature, we can see that the energy along Volmer-Tafel reaction pathway (marked by black V-T line) is larger than that along Volmer-Heyrovsky (marked by red V-H) process, indicating that Volmer-Heyrovsky mechanism has obvious advantages in HER process. The detailed discussion has been provided in the part of “proposed HER mechanism” in pages 11-12.

In addition, according to the Reviewer 3’s suggestion and listing reference [J. Am. Chem. Soc. 2015, 137, 7365-7370 and Nat. Commun. 2018, 9, 1425], our Tafel slope observed should be attributed to Volmer-Heyrovsky mechanism, consistent with our current calculations.

Figure R2-10. Schematic configuration-coordinate diagrams for the Volmer-Heyrovsky (V-H) and the Volmer-Tafel (V-T) HER mechanism.

Free energy versus the reaction coordinates of HER for different active sites. (Fe, gray spheres, S, pink spheres, H, blue spheres).

The corresponding descriptions have been revised and marked by red, see pages 11-12 lines 240-264.

Comment 23. 12) *What is the amount of FeS deposited on the CFC? Could the authors provide a “intrinsic” activity of the catalysts, that is the current density per mg of catalyst at a given over potential (for instance $\eta = -0.1V$)?*

Answer: Thanks for the reviewer’s suggestion. The analysis about digital photographs and schematic illustration of the CFC and the FeS/CFC electrode (Figure R2-11) shows that the amount of FeS deposited on CFC can be obtained by the quantity difference between CFC [m(CFC)=34.8 mg] and FeS/CFC [m(FeS/CFC)=35.2 mg] . The surface area of CFC is about S=2.5 cm². Then amount of FeS can be calculated to be

$$\varnothing = \frac{m(\text{FeS/CFC}) - m(\text{CFC})}{S} = 0.16 \text{ mg/cm}^2$$

The current density per mg is 7.13 mA/mg at over potential of -0.1 V.

Figure R2-11. Digital photographs (top) and schematic illustration (bottom) of CFC (a) and FeS/CFC (b) electrodes

The corresponding descriptions have been added into page S10 and page 4 lines 71-72, this figure is renamed as Figure S2.

Comment 24. 13) *The control figure S9 is not convincing and should be improved by adding the curves corresponding to the same experiments but performed at 328.4 K to evidence the absence of change.*

Answer: Thanks for the reviewer's suggestion. The same experiments performed at 328.4 K in Figure R2-12 evidence that there are no obvious changes.

Figure R2-12. I-V curves of the photodetector with and without NIR light (1064 nm) illumination at 336.9 K (a) and 328.4 K (b).

These results have been added into Fig. S12 in page S21.

Comment 25. 14) *Values of the fits for impedance spectroscopy (fig. 4g) should be presented in a table in SI, especially the changes in RCT values.*

Answer: Thanks for the reviewer's suggestion. We have fitted impedance

spectroscopy to evaluate the values of R_s and R_{ct} with and without NIR light radiation, and the acquired values are presented in Table R2-3. The R_{ct} value at 298 K is slightly reduced from 41.51 to 35.94 Ω due to NIR irradiation, but that value at 323 K decreases nearly by half. Associated with substantially theoretical and experimental results in this manuscript, we conclude that the photoinduced semiconductor-metal phase transition is responsible for this significant decrease of R_{ct} value at 323 K.

Table R2-3. Results of fitting Nyquist plots by equivalent circuit.

Sample	R_s (Ω)	R_{ct} (Ω)
323 K Light	1.15	13.12
323 K Dark	1.17	30.62
298 K Light	1.18	35.94
298 K Dark	1.21	41.51

These results have been added into page 11 line 231 and page S34, renamed as Table S3.

End

Response to the report of the reviewer 3

Comment 1. *The authors have realized the semiconductor-metal transition of troilite FeS by using near infrared radiation. Remarkably, the resulting metallic phase exhibits higher catalytic activity than its semiconductor counterpart. Below my comments and suggestions:*

Question: The value U of the Hubbard parameter

Answer: Thanks for the reviewer's comments and suggestion. To better understand the semiconductor-metal phase transition, the DFT+U correction method is adopted to treat the localized d orbits of Fe atom. After analysis and comparison, we find that the GGA-PBE exchange-correlation function with $U=1$ eV gives the lattice constants ($a=3.44$ Å, $c=5.85$ Å for $P6_3/mmc$ phase; $a=5.97$ Å, $c=11.73$ Å for $P\bar{6}2_c$ phase) in perfect agreement with experiments [Acta. Chemica Scandinavica, 14, 919 (1960)].

This revised part has been inserted into page S2.

Comment 2. *Which among the three terminations (J. Phys. Chem. C 122, 24, 12810-12818) of the (100) prismatic surface of troilite has been used in the calculations.*

Answer: Similar to previous reports [J. Phys. Chem. C, 2018, 122, 12810], there are three terminations, such as Fe-termination, S- termination, or Fe and S termination. In our calculation, the (100) surface with Fe and S terminations is considered by expanding the slab symmetrically around the reflection plane. This is because the basal surface with Fe termination is polar, which is unstable to realize good catalytic stability. The surface model with S-terminations cannot provide high catalytic activity to explain our experiment results.

This revised part has been inserted into page S2, and this reference is added as Ref 28.

Comment 3. *The thickness of the surface slab in terms of FeS layers.*

Answer: After implementing convergence test, in our calculation, the FeS (100) surface is constructed as (1×2) supercell consisting of a six-trilayer slab and a vacuum with thickness of 20 Å to avoid the interaction between periodical images. The bottom two trilayers are fixed to mimic the bulk.

This revised part has been inserted into page S2.

Comment 4. *The bulk lattice vectors used to construct the surface unit cells.*

Answer: According to crystal symmetry, the lattice vectors along (010) and (001) direction are used to construct the (100) surface.

This revised part has been inserted into page S2.

Comment 5. *Please note that the convergence of the plane wave cutoff and k-point grid is not related to the convergence of the forces on the ions, which seems to be*

within 0.03 eV/Å. Thus, the expression "in order to achieve a satisfactory degree of convergence (0.03 eV/Å)" should be avoided and the ion forces threshold of 0.03 eV/Å simply listed.

Answer: Thanks for the reviewer's comments and suggestion. The description about “We used projected augmented-wave (PAW) pseudopotentials, and in order to achieve a satisfactory degree of convergence (0.03 eV/Å) the plane wave expansion has been truncated at a cutoff energy of 550 eV” has been revised as “We used projected augmented-wave (PAW) pseudopotentials, and in order to achieve a satisfactory degree of convergence the plane wave expansion has been truncated at a cutoff energy of 550 eV”. The description about “free ions” are changed as “the relaxation is carried out until all forces on the free ions are converged to 0.03 eV/ Å”

This revised part has been inserted into page S2.

Comment 6. *FeS nanoparticles have already been reported to exhibit molecular hydrogen evolution in neutral water at room temperature (ACS Catal. 4, 2, 681-687). The authors should acknowledge the study above and try to compare the values of overpotential, exchange current density and Tafel slope reported there to those of the room temperature semiconductor FeS phase of this work.*

Answer: Thanks for the reviewer's comments and suggestion. We have compared our results with previous reports, as shown in the following Table R3-1, and the relevant references have been added into our manuscript. The comparison results indicate that our synthesized FeS catalysts have the lowest Tafel slope and excellent stability. The reference (ACS Catal. 4, 2 681-687) has been added into our revised manuscript

as Ref. 16.

Table R3-1. The comparison of HER performance between FeS/CFC and previously reported Fe-based electrocatalysts.

Catalyst	Electrolyte solution	Overpotential (mV vs. RHE)	Exchange current density (j_0 , $\mu\text{A cm}^{-2}$)	Tafel slope (mV per decade)	Stability test	Reference
FeS/CFC	0.1 M KOH	142	3.1	36.9	100 h	This work
FeS ₂ /CoS ₂	1.0 M KOH	78.2	–	44	80 h	Small, 2018, 14, 1801070
A-FeNiS	0.5 M H ₂ SO ₄	105	2.2	40	40 h	J. Am. Chem. Soc. 2015, 137, 11900–11903
NiFe/NiCo ₂ O ₄ /NF	1.0 M KOH	105	470	88	10 h	Adv. Funct. Mater. 2016, 26, 3515
Ni-Fe/NC	1.0 M KOH	219	–	110	1200 s	ACS Catal. 2016, 6, 580
Meso-FeS ₂	0.1 M KOH	96	630	78	24 h	J. Am. Chem. Soc. 2017, 139, 13604
pyrite FeS ₂ film	0.5 M H ₂ SO ₄	270	–	62.5	–	Energy Environ. Sci., 2013, 6, 3553
Pyrite FeS ₂ /C	0.1 M phosphate buffer	557	–	204	–	ACS Catal. 2016, 6, 2626
Ni Fe LDH-NS DG10	1.0 M KOH	300	–	110	20000 s	Adv. Mater. 2017, 29, 1700017
FeS ₂ @MoS ₂ /rGO	0.5M H ₂ SO ₄	123	17.5	38.4	8 h	Chem. Commun., 2016, 52, 11795
FeS P	0.5M H ₂ SO ₄	160	–	44	–	ACS Catal. 2017, 7, 4026
2-cycle NiFeO _x /CFP	1.0 M KOH	88	–	150	100 h	Nature Comm. 2015, 6, 7261
FeS	1.0 M Phosphate buffer	350	0.66	150	–	ACS Catal. 2014, 4, 681
FeCoNi-HNT	1.0 M KOH	58	–	37.5	100 h	Nature Comm. 2018, 9, 2452
(Fe _{0.48} Co _{0.52})S ₂	0.5 M H ₂ SO ₄	196	0.959	47.5	20 h	J. Phys. Chem. C 2014, 118, 21347.
Fe-CoP/Ti	1.0 M KOH	78	–	75	20 h	Adv. Mater. 2017, 29, 1602441
Fe-N-C	0.5 M H ₂ SO ₄	130	27	89	5 h	Adv. Energy Mater. 2018, 8, 1701345
Ni-Fe-P porous nanorods	1.0 M KOH	79	–	92.6	24 h	J. Mater. Chem. A, 2017, 5, 2496
Co _{0.6} Fe _{0.4} P/C NT	0.5 M H ₂ SO ₄	67	–	57	24 h	Adv. Funct. Mater. 2017, 27, 1606635
FeP NAS/CC	1.0 M KOH	218	–	146	20 h	ACS Catal. 2014, 4, 4065

These parts have been added into page 10 lines 216-218, and renamed as Table S2 in

page S33.

Comment 7. *The authors have indicated that the phase at high temperature is hexagonal ($P6_3/mmc$). However, while most of the works point towards that direction (*J. Phys. Chem. Solids* 2017, 111, 317-323 or *Science* 1995, 268, 1892), the appearance of a phase with an orthorhombic cell ($Pnma$) has also been suggested (*J. Solid State Chem.* 1990, 84, 194-210 or *Am. Mineral.* 1992, 77, 391-398). Do the authors have data supporting the $P6_3/mmc$ phase? Can they comment on this point?*

Answer: Thanks for the reviewer's comments and suggestion. In order to identify the structural transition, the XRD patterns at temperatures of 298 K and 403 K were measured and shown in Figure R3-1. We can see that the XRD peaks are up-shifted slightly, accompanied by phase transition from $P\bar{6}2_c$ to $P6_3/mmc$, which is quite in agreement with our theoretical simulation. In addition, the XRD feature of $Pnma$ phase is obviously different from our experimental results. Therefore, the comparison of XRD fingerprint further confirms that the phase transition occurs between $P\bar{6}2_c$ and $P6_3/mmc$.

[Redacted]

Comment 8. *The title indicates a reversibility of the transition. Can the authors discuss it somewhere also in the manuscript?*

Answer: Thanks for the reviewer's comments and suggestion. In fact, the changes in

Raman spectra have demonstrate the reversibility of transition. The NIR light irradiation make E_g mode at 323 K down-shift to the same position as that at 403 K without NIR light, as shown in Figure R3-2, which indicates that the phase transition temperature of ultrathin FeS nanosheets decreases to 323 K. When the NIR light radiation is removed, the E_g mode at 323 K is up-shifted slightly and tends to the peak position at 323 K without NIR light radiation. The changes in Raman spectra induced by NIR light radiation successfully confirm the reversibility of the transition. Some necessary discussion has been inserted into our manuscript, see page 6 lines 116-117.

In addition, this reversibility can make catalyst restore to initial state easily, different from traditional method such as constructing various nanostructures, introducing vacancies or dopants and so on. This can be used to design new-type memorizing catalyst. However, we focus on the improvement of HER performance by a simple photoinduced semiconductor-metal phase transition in this work. In order to highlight the theme and simultaneously consider the suggestion of the Reviewer 2, the emphasized description about “reversibility of this transition” in title, abstract and conclusion has been removed partially.

Figure R3-2. The Raman spectra of ultrathin FeS nanosheets acquired at different temperatures.

Comment 9. Line 56: *"Herein, for the first time, we realize the photoinduced reversible semiconductor-metal transition at near room temperature in ultrathin troilite FeS nanosheets vertically grown on carbon fiber cloth."* At this point of the manuscript, the general reader may not know what exactly should be considered as novel in this work ("for the first time"). The reversibility? The transition? The room temperature? The nanosheets? The carbon fiber cloth substrate? A combination of them? Can the authors clarify? The abstract should be changed as well.

Answer: Thanks for the reviewer's comments and suggestion. In order to avoid misunderstanding, this sentence has been rewritten as "Herein, we report the highly efficient hydrogen evolution triggered by photoinduced semiconductor-metal transition at near room temperature for the first time. This phase transition occurs in

ultrathin troilite FeS nanosheets vertically grown on carbon fiber cloth (CFC)."

Besides, the corresponding sentence in the abstract has been revised as well. See page 3 lines 55-58.

Comment 10. *Line 87: "which is in consistence with the theoretical simulation" should be replaced by "which is in consistence with the structure of troilite"*

Answer: This unsuitable description has been revised according to the reviewer's suggestion, please see page 4 line 88.

Comment 11. *Line 180: "From the difference in charge density of the two phases in upper panel in Fig. 4b, we find the change only happens in charge density localized on Fe atoms (red region), showing Fe atoms has higher catalytic activity." Can the authors clarify this sentence? It is not clear to me why the change in charge density should give a higher catalytic activity.*

Answer: It is generally known that a material displays a good catalytic activity when the free energy of adsorbed hydrogen tends to be thermoneutral, i.e., $\Delta G_{\text{H}}=0$. If hydrogen cannot efficiently adsorb onto catalyst or form a strong chemical bond, the hydrogen release and proton/electron transfer process will be limited, leading to lower catalytic activity. However, the adsorption of H^* onto Fe atoms strongly depends on their difference in Pauling electronegativity. The change in charge density induced by structural phase transition mainly occurs at surficial Fe atom, leading to an increase of

difference in Pauling electronegativity between H^* and Fe from 0.23 ($P\bar{6}2_c$ phase) to 0.37 ($P6_3/mmc$ phase). Therefore, the Gibbs free energy of hydrogen adsorption tends to zero (as shown in Fig. 4a), which is benefit for improving catalytic activity.

Some suitable discussion has been inserted into page 9 lines 183-193.

Comment 12. *Line 183: "This agrees with Fig. 4a because more electrons are transferred into these Fe sites to optimize the adsorption of hydrogen (Fig. S12)." What I see in Figure 4a is that with the phase transition the Gibbs free energy of the adsorbed H on Fe1 (Fe2) increases (decreases) to zero (which is beneficial), but I do not understand how this correlates with Fig. S12, which shows that both Fe1 and Fe2 gain charge. In addition, I am a bit surprised by the fact that Fe1 and Fe2 give opposite signs for the adsorption energy of H. Can the authors comment on this? Please note that it is not easy to visualize the different arrangement of Fe1 and Fe2 from the upper panel of Fig. 4b. Finally, I do not believe that the results on the charge transfer, i.e. the upper panel of Fig. 4b and Fig. S12 strengthen the manuscript and therefore they may be removed, while, a better figure with the Fe1, Fe2 and S the labels should be added.*

Answer: Thanks for the reviewer's comments and suggestion. As discussed in above comment 11, the Gibbs free energy of the adsorbed H is strongly interrelated with the difference in Pauling electronegativity between H^* and Fe, which can be affected by charge transfer. In order to avoid misunderstanding, the discussion about charge transfer in Figure S12 has been removed.

From the analysis about its crystal symmetry, the Fe1 and Fe2 of $P\bar{6}2_c$ phase surface are distributed at two sides of symmetric axis as shown in Figure R3-3(a).

When the structural phase transits from $P\bar{6}2_c$ to $P6_3/mmc$, the Fe1 and Fe2 atoms are distorted to symmetric axis ($P6_3/mmc$, Figure R3-3b) from opposite direction (marked by black arrow), finally tend to syequilibrium position (marked by purple line). Therefore, the Gibbs free energy of adsorption at Fe1 and Fe2 sites demonstrates opposite signs.

Figure R3-3. The atomic distortion process from $P\bar{6}2_c$ phase (a) to $P6_3/mmc$ phase (b).

According to the reviewer's suggestion, the charge transfer in upper panel of Fig. 4b and Fig. S12 have been removed, and the more intuitive Figure with labeled Fe1, Fe2 and S is provided as following Figure R3-4b.

Figure R3-4. Practical application of HER. (a) The Gibbs free energy as a function of the fraction of the atomic distortion between the $P\bar{6}2_c$ (0%) and $P6_3/mmc$ (100%) phase. (b) Top panel: Atomic structural model of FeS (100) surface. Bottom panel: DOS of superficial Fe and S atoms.

Comment 13. Line 184: "The atomic density of state (DOS) analysis confirms that the highest occupied orbital is composed by not p -states of S atoms but d -states of Fe atoms (lower panel in Fig. 4b), which is benefit for optimizing the electronic structures to improve HER performance (see detailed orbital resolved DOSs in Figs. S13 and S14)." Can the authors clarify this argument?

Answer: Thanks for the reviewer's comments and suggestion. From the atomic density of state (DOS) analysis in Fig. 4b, we can see that the highest occupied orbit is not composed by p -states of S atoms but d -states of Fe, which is quite in agreement with detailed orbital resolved DOS in Fig. S15 and S16. The transition from semiconductor to metal phase (see analysis about DOS in Fig. S15 and S16) will improve carrier transfer and benefit for HER performance. In order to better

understand our argument, this sentence is revised as “The atomic density of state (DOS) analysis confirms that the highest occupied orbit is not composed by p-states of S atoms but d-states of Fe atoms (lower panel in Fig. 4b), and the corresponding electronic structure transition from semiconductor to metal phase (see detailed orbital resolved DOSs in Figs. S15 and S16) will benefit for improving carrier transfer and HER performance.”

Some suitable discussion has been inserted into page 9 lines 183-193.

Comment 14. *Line 202: "Moreover, such low Tafel slope suggests that Volmer Tafel mechanism plays a predominant role in determining the HER rate and the recombination step is the rate-limiting step". I believe that the Tafel slopes of the curves of Fig. 4d may suggest instead a Volmer-Heyrovsky mechanism (J. Am. Chem. Soc. 2015, 137, 7365-7370 or J. Mater. Chem. A, 2015, 3, 1494). Why has it been excluded?*

Answer: Thanks for the reviewer's comments and suggestion. After carefully reading these references [J. Am. Chem. Soc. 2015, 137, 7365-7370 and Nat. Commun., 2018, 9, 1425], we completely agree with the reviewer's suggestion that those Tafel slopes in Fig. 4d can be ascribed to Volmer-Heyrovsky mechanism. In order to clarify the HER mechanism, the relative energy profiles along the reaction pathway with Volmer-Heyrovsky and Volmer-Tafel process are recalculated and compared, as shown in Figure R3-5. The calculated result demonstrates that the energy along Volmer-Tafel reaction pathway (marked by black V-T line) is larger than that along Volmer-Heyrovsky (marked by red V-H) process, indicating that Volmer-Heyrovsky

mechanism has obvious advantage in HER process.

Figure R3-5. Schematic configuration-coordinate diagrams for the Volmer-Heyrovsky (V-H) and the Volmer-Tafel (V-T) HER mechanism. Free energy versus the reaction coordinates of HER for different active sites. (Fe, gray spheres, S, pink spheres, H, blue spheres).

This sentence has been revised as “Moreover, such low Tafel slope suggests that Volmer-Heyrovsky mechanism plays a predominant role in determining the HER rate and the electrochemical desorption step is the rate-limiting step”. Those relevant references have been added as refs 7 31, 32, 36. In addition, some corresponding description has also been revised, please see page 11-12, lines 240-264.

Comment 15. *"The energy along semiconductor phase (P62c) reaction pathway (marked by red line) is larger than that along metal phase (P6₃/mmc) (marked by black line), indicating that metal phase has obvious advantages in HER process." This sentence has little meaning in my opinion. From Fig. 5, it is hard to infer why the*

efficiency of the metal phase should be higher than that of the semiconductor, as the activation barriers look very similar. I suggest the authors to remove this sentence from the manuscript and to use Fig. 5 only to speculate a possible Volmer-Tafel pathway for the two phases. However, the authors should compare it with that of an alternative Volmer Heyrovsky mechanism. Please see my comment above.

Answer: Thanks for the reviewer's comments and suggestion. In order to clarify the HER mechanism, the relative energy profiles along the reaction pathway with Volmer-Heyrovsky and Volmer-Tafel process are recalculated for a comparison, as shown in Figure R3-5.

These descriptions have been added into page 11-12, lines 240-264.

Comment 16. *Line 250: "Therefore, the semiconductor-metal phase transition plays a critical role in improving HER performance". I do not think that this conclusion is supported by Fig. 5, and therefore it should be removed from the text.*

Answer: Thanks for the reviewer's comments and suggestion. This inappropriate sentence has been removed.

End

Reviewers' comments:

Reviewer #1 (Remarks to the Author):

The authors have satisfactorily answered my comments.
I recommend the paper for publication without further change.

Reviewer #2 (Remarks to the Author):

Review on revised version of the manuscript NCOMMS-18-27543A

I acknowledge the efforts the authors have made to significantly improve the first draft of the paper. Many questions and suggestions have been answered and the quality of the paper is much better now.

However I have some remarks about the answers about some of my comments which I will detail below.

Comment 11. The table is very informative but still lacking some data. The stability of the FeS-based compounds in Energy Environ. Sci., 6, 3553-3558 (2013) is ca. 4000 s, those in ACS Catal. 4, 681-687 (2014) and ACS Catal. 6, 2626-2631 (2016) are stable for more than 120 h and the FeS/P-compounds reported in ACS Catal. 7, 4026-4032 (2017) have a stability over 2h. This must be added in the table.

Comment 12. I agree with fact that introducing Ni or Co elements in iron sulfides enhance the catalytic activity but will the SC to metal transition remains in such new systems?

Comment 20. I thank the authors for their answer and for adding the the XPS experiments. When I said "polyol could induce sulfate moieties at the surface", I forgot the thermal treatment under H_2 . Could the authors provide XPS spectrum of the sample before the thermal treatment?

Anyway, I think that those minor points below answered, the paper has gained in clarity and quality. Although I still have some doubts about the practical application of such materials, I reckon that the underlying idea is interesting and that the paper is suitable for publication in Nat. Commun.

One remark, I know it is too late to ask for complementary experiments right now, but I am always surprised that, when Fe-based materials are concerned and especially this phase transition between semi-conducting and metallic phases, no Mössbauer experiments have been carried out. In principle, they should have evidenced with no doubt the phase transition.

Reviewer #3 (Remarks to the Author):

The authors have improved the computational part of the manuscript. I am happy with most of the points raised in the first round, but I feel that they should still address the following ones.

Comment 1

After analysis and comparison, we find that the GGA-PBE exchange correlation function with $U=1$ eV gives the lattice constants ($a=3.44$ Å, $c=5.85$ Å for P63/mmc phase; $a=5.97$ Å, $c=11.73$ Å for P62c phase) in perfect agreement with experiments [Acta. Chemica Scandinavica, 14, 919 (1960)].

Please mention PBE in the Methods. The value of U is identical to that used in previous works on troilite (Phys. Rev. Lett. 2016, 116, 227601 and J. Phys. Chem. Solids 2017, 111, 317-323). In the absence of additional reported data on the structural properties of FeS as a function of U , I suggest the authors to provide those two references as a more solid justification for their choice of the Hubbard parameter.

Comment 2

Similar to previous reports [J. Phys. Chem. C, 2018, 122, 12810], there are three terminations, such as Fe-termination, S- termination, or Fe and S termination. In our calculation, the (100) surface with Fe and S terminations is considered by expanding the slab symmetrically around the reflection plane. This is because the basal surface with Fe termination is polar, which is unstable to realize good catalytic stability. The surface model with S-terminations cannot provide high catalytic activity to explain our experiment results.

This is very confusing. The authors state more times in the manuscript that the (100) surface refers to the room temperature P6-2c troilite phase. However, from the reply above I am now fully convinced that is that of the P63/mmc phase. They should clarify this point and mention the correct P63/mmc phase (e.g. in lines 32 and 437 among others of SI, and lines 242 and 441 of the manuscript).

Comment 4

According to crystal symmetry, the lattice vectors along (010) and (001) direction are used to construct the (100) surface.

This becomes superfluous once the authors mention in the text the values of the lattice parameters and clarify the phase (see comment 2).

Comment 6

The reference (ACS Catal. 4, 2 681-687) has been added into our revised manuscript as Ref. 16.

Please add "sulfides" among the materials of line 43 for consistency with the added reference.

Comment 11

However, the adsorption of H^* onto Fe atoms strongly depends on their difference in Pauling electronegativity. The change in charge density induced by structural phase transition mainly occurs at surficial Fe atom, leading to an increase of difference in Pauling electronegativity between H^* and Fe from 0.23 (P62c phase) to 0.37 (P63/mmc phase). Therefore, the Gibbs free energy of hydrogen adsorption tends to zero (as shown in Fig. 4a), which is benefit for improving catalytic activity.

The authors have introduced the concept of Pauling electronegativity at this point. I am not convinced by their explanation, as the electronegativity should increase with the strength of the binding energy. They report instead an opposite trend. I believe that lines 183-188 should better be removed, especially because the results on the charge transfer of the upper panel of Fig. 4b are not there any

more.

Comment 12

From the analysis about its crystal symmetry, the Fe1 and Fe2 of P62c phase surface are distributed at two sides of symmetric axis as shown in Figure R3-3(a). When the structural phase transits from P62c to P63/mmc, the Fe1 and Fe2 atoms are distorted to symmetric axis (P63/mmc, Figure R3-3b) from opposite direction (marked by black arrow), finally tend to syequilibrium position (marked by purple line). Therefore, the Gibbs free energy of adsorption at Fe1 and Fe2 sites demonstrates opposite signs.

I do not agree with the authors on this point. It is unlikely that a different coordinate along the surface as that reported in Figure R3-3(a) would result in opposite adsorption energies. The difference between Fe1 and Fe2 may instead be due to their different positions along the normal to the surface, which is not possible to appreciate either from figure R3-3 or figure 4b. Can the authors look into it?

Comment 13

The atomic density of state (DOS) analysis confirms that the highest occupied orbit is not composed by p-states of S atoms but d-states of Fe atoms (lower panel in Fig. 4b), and the corresponding electronic structure transition from semiconductor to metal phase (see detailed orbital resolved DOSs in Figs. S15 and S16) will benefit for improving carrier transfer and HER performance.

Is it important for the main message of the work that the highest occupied orbitals are composed by d-states of Fe atoms? Once showing (line 99) in figure 2b (perhaps figure S16 could be also referenced in line 99) the appearance of a band gap, do the lower panel of figure 4b, figure S15 and lines 188-193) contribute to make stronger the manuscript? Personally I think that they can be removed.

Comment 15

The calculated result demonstrates that the energy along Volmer-Tafel reaction pathway (marked by black V-T line) is larger than that along Volmer-Heyrovsky (marked by red V-H) process, indicating that Volmer-Heyrovsky mechanism has obvious advantage in HER process.

I appreciate that the authors have now included the Volmer-Heyrovsky mechanism. From the new comparison, it is again difficult to say which of the two mechanisms will occur, as both profiles present high activation barriers. Given this circumstance, I suggest the authors to use Fig. 5 only to present a possible pathway of the mechanism suggested by the experimental data, i.e. the Volmer-Heyrovsky (the phase used in these calculations should be mentioned). Lines 241-263 should be modified accordingly. Finally, the sentence "Therefore, the Volmer-Heyrovsky mechanism plays a critical role in improving HER performance." should be deleted as I do not think that it is currently adequately supported by Fig. 5.

Response to the report of the reviewer 2

Comment 1. *Review on revised version of the manuscript NCOMMS-18-27543A. I acknowledge the efforts the authors have made to significantly improve the first draft of the paper. Many questions and suggestions have been answered and the quality of the paper is much better now. However I have some remarks about the answers about some of my comments which I will detail below.*

Question: Comment 11. The table is very informative but still lacking some data. The stability of the FeS-based compounds in Energy Environ. Sci., 6, 3553-3558 (2013) is ca. 4000 s, those in ACS Catal. 4, 681-687 (2014) and ACS Catal. 6, 2626-2631 (2016) are stable for more than 120 h and the FeS/P-compounds reported in ACS Catal. 7, 4026-4032 (2017) have a stability over 2h. This must be added in the table.

Answer: Thanks for the reviewer's comments and suggestions. Those information about stability test have been added into Table S2, as follows:

Table S2. The comparison of HER performance between FeS/CFC and previously reported Fe-based electrocatalysts.

Catalyst	Electrolyte solution	Overpotential (mV vs. RHE)	Exchange current density (j_0 , $\mu\text{A cm}^{-2}$)	Tafel slope (mV per decade)	Stability test	Reference
FeS/CFC	0.1 M KOH	142	3.1	36.9	100 h	This work
FeS ₂ /CoS ₂	1.0 M KOH	78.2	–	44	80 h	Small, 2018, 14, 1801070
A-FeNiS	0.5 M H ₂ SO ₄	105	2.2	40	40 h	J. Am. Chem. Soc. 2015, 137, 11900
NiFe/NiCo ₂ O ₄ /NF	1.0 M KOH	105	470	88	10 h	Adv. Funct. Mater. 2016, 26, 3515
Ni-Fe/NC	1.0 M KOH	219	–	110	1200 s	ACS Catal. 2016, 6, 580
Meso-FeS ₂	0.1 M KOH	96	630	78	24 h	J. Am. Chem. Soc. 2017, 139, 13604
pyrite FeS ₂ film	0.5 M H ₂ SO ₄	270	–	62.5	4000 s	Energy Environ. Sci., 2013, 6, 3553
Pyrite FeS ₂ /C	0.1 M phosphate buffer	557	–	204	120 h	ACS Catal. 2016, 6, 2626
Ni Fe LDH-NS DG10	1.0 M KOH	300	–	110	20000 s	Adv. Mater. 2017, 29, 1700017
FeS ₂ @MoS ₂ /rGO	0.5M H ₂ SO ₄	123	17.5	38.4	8 h	Chem. Commun., 2016, 52, 11795
FeS P	0.5M H ₂ SO ₄	160	–	44	2 h	ACS Catal. 2017, 7, 4026
2-cycle NiFeO _x /CFP	1.0 M KOH	88	–	150	100 h	Nature Comm. 2015, 6, 7261
FeS	1.0 M Phosphate buffer	350	0.66	150	120 h	ACS Catal. 2014, 4, 681
FeCoNi-HNT	1.0 M KOH	58	–	37.5	100 h	Nature Comm. 2018, 9, 2452
(Fe _{0.48} Co _{0.52})S ₂	0.5 M H ₂ SO ₄	196	0.959	47.5	20 h	J. Phys. Chem. C 2014, 118, 21347.
Fe-CoP/Ti	1.0 M KOH	78	–	75	20 h	Adv. Mater. 2017, 29, 1602441
Fe-N-C	0.5 M H ₂ SO ₄	130	27	89	5 h	Adv. Energy Mater. 2018, 8, 1701345
Ni-Fe-P porous nanorods	1.0 M KOH	79	–	92.6	24 h	J. Mater. Chem. A, 2017, 5, 2496
Co _{0.6} Fe _{0.4} P/C NT	0.5 M H ₂ SO ₄	67	–	57	24 h	Adv. Funct. Mater. 2017, 27, 1606635
FeP NAS/CC	1.0 M KOH	218	–	146	20 h	ACS Catal. 2014, 4, 4065

Comment 2. *Comment 12. I agree with fact that introducing Ni or Co elements in iron sulfides enhance the catalytic activity but will the SC to metal transition remains*

in such new systems?

Answer: Thanks for the reviewer's comments and reminding. In fact, the investigation on doping Ni and Co into our sample is being considered as the next work. Our preliminary results indicate that the transition from semiconductor to metal phase strongly depends on doping concentration. When the doping concentration of Co or Ni element is too high, this phase transition may be depressed. The dependence of the optimal HER performance on dopants and phase transition will be discussed in detail in next work.

Comment 3. *Comment 20. I thank the authors for their answer and for adding the XPS experiments. When I said "polyol could induce sulfate moieties at the surface", I forgot the thermal treatment under H₂S. Could the authors provide XPS spectrum of the sample before the thermal treatment?*

Answer: Thanks for the reviewer's comments and reminding. The XPS spectrum of the sample before the thermal treatment is shown in Figure R2-1. The fine XPS spectra of Fe 2p (Figure R2-1a) and S 2p (Figure R2-1b) indicate that the binding energies of Fe 2p_{1/2}, Fe 2p_{3/2}, S 2p_{1/2} and S 2p_{3/2} are located at 721.3, 708.1, 163.8 and 161.9 eV, respectively. In this unannealed sample, a weak XPS peak at 169.5 eV originating from surface sulfate moieties (SO₄²⁻) can be observed due to the polyol synthesis progress of metal sulfides. In comparison with the annealed sample in Fig. S20, this weak XPS peak originating from SO₄²⁻ can be removed. It is important to note that the characteristic peaks of Fe 2p shift toward higher binding energy after thermal treatment in H₂S atmosphere, which can be ascribed to the decreased electron

density induced by disappearance of iron deficiencies. These results can also be confirmed by XRD in Fig. S4.

Figure R2-1. XPS spectra of FeS nanosheets before thermal treatment.

This figure has been renamed as Figure S5, and the descriptions have been added into page S13 and page 4.

Comment 4. *Anyway, I think that those minor points below answered, the paper has gained in clarity and quality. Although I still have some doubts about the practical application of such materials, I reckon that the underlying idea is interesting and that the paper is suitable for publication in Nat. Commun.*

One remark, I know it is too late to ask for complementary experiments right now, but I am always surprised that, when Fe-based materials are concerned and especially this phase transition between semi-conducting and metallic phases, no Mössbauer experiments have been carried out. In principle, they should have evidenced with no doubt the phase transition.

Answer: Thanks for the reviewer's comments and kindness. To provide more substantial experimental evidences, ^{57}Fe Mössbauer spectra were acquired from FeS

sample with and without NIR light irradiation, as shown in Figure R2-2(a). The schematic illustration of nuclear Zeeman splitting in Figure R2-2(b) indicates that hyperfine interaction induced by electric quadrupole and magnetic dipole leads to six typical resonance absorption, consistent with our expected Mössbauer spectra. The obtained values of the isomer shift (IS=0.83) and quadrupole splitting (QS=-0.15) without NIR light irradiation are comparable to previous data in the literature [Ref.S8 and Ref. S9]. When the NIR light is introduced, the more symmetrical spectral fingerprint and the changed IS (0.79) and QS (-0.12) values are induced by reconfiguration of charge density accompanied by structural transformation. The analysis about atomic density of state (DOS) confirms that the highest occupied orbit is not composed by p-states of S atoms but d-states of Fe atoms (Figure S7). And the corresponding electronic structure transition from semiconductor to metal phase will lead to an effective electron wave function rearrangement of d orbits, finally affecting electron density of s orbit nearby nucleus. By comparing the electron localization functions of two phases in Figure R2-2c and R2-2d, we also found that the electrons have smaller tendency to locate around pristine atoms in metallic $P6_3/mmc$ phase (Fig. R2-2c) than those in semiconductor $P\bar{6}2c$ phase (Fig. R2-2d), which can be used to explain the change in IS and QS value induced by phase transition. The changes in both Mössbauer spectra and corresponding analysis about electronic structure confirm that the phase transition can occur in our sample, which is quite in agreement with our conclusion associated with Raman spectra, photo-electricity response and comprehensive theoretical analysis.

Figure R2-2. (a) ^{57}Fe Mössbauer spectra of FeS nanosheets with and without NIR light irradiation. (b) The schematic illustration of nuclear Zeeman splitting. The calculated electron localization function of the (100) surface of $P\bar{6}2_c$ (c) and $P6_3/mmc$ (d) phase.

This figure has been renamed as Figure S10, and the descriptions have been added into page S18-S19 and page 6.

End

Response to the report of the reviewer 3

Comment 1. *The authors have improved the computational part of the manuscript. I am happy with most of the points raised in the first round, but I feel that they should still address the following ones.*

Comment 1: After analysis and comparison, we find that the GGA-PBE exchange correlation function with $U=1$ eV gives the lattice constants ($a=3.44$ Å, $c=5.85$ Å for P63/mmc phase; $a=5.97$ Å, $c=11.73$ Å for P62c phase) in perfect agreement with experiments [Acta. Chemica Scandinavica, 14, 919 (1960)].

Please mention PBE in the Methods. The value of U is identical to that used in previous works on troilite (Phys. Rev. Lett. 2016, 116, 227601 and J. Phys. Chem. Solids 2017, 111, 317-323). In the absence of additional reported data on the structural properties of FeS as a function of U , I suggest the authors to provide those two references as a more solid justification for their choice of the Hubbard parameter.

Answer: Thanks for the reviewer's comments and suggestions. The GGA-PBE exchange correlation function means “generalized gradient approximation (GGA) and Perdew-Burke-Ernzerof (PBE) exchange-correlation functional”, which has been added into page S2. In addition, two relevant references have also been added as Ref. S3 and S4.

Comment 2. *Similar to previous reports [J. Phys. Chem. C, 2018, 122, 12810], there are three terminations, such as Fe-termination, S- termination, or Fe and S termination. In our calculation, the (100) surface with Fe and S terminations is considered by expanding the slab symmetrically around the reflection plane. This is*

because the basal surface with Fe termination is polar, which is unstable to realize good catalytic stability. The surface model with S-terminations cannot provide high catalytic activity to explain our experiment results.

This is very confusing. The authors state more times in the manuscript that the (100) surface refers to the room temperature P62c troilite phase. However, from the reply above I am now fully convinced that is that of the P63/mmc phase. They should clarify this point and mention the correct P63/mmc phase (e.g. in lines 32 and 437 among others of SI, and lines 242 and 441 of the manuscript).

Answer: Thanks for the reviewer's comments and suggestions. As convinced by the reviewer, the (100) surface in manuscript are mainly constructed from $P6_3/mmc$ phase. This point has been clarified in revised manuscript. In order to eliminate confusion, some supplementary descriptions have been added into revised manuscript.

Comment 3. *According to crystal symmetry, the lattice vectors along (010) and (001) direction are used to construct the (100) surface. This becomes superfluous once the authors mention in the text the values of the lattice parameters and clarify the phase (see comment 2).*

Answer: Thanks for the reviewer's comments and suggestions. The phase structure at (100) surface has been clarified as comment 2. Besides, the superfluous sentence about lattice vectors has been removed.

Comment 4. *The reference (ACS Catal. 4, 2 681-687) has been added into our revised manuscript as Ref. 16. Please add "sulfides" among the materials of line 43*

for consistency with the added reference.

Answer: Thanks for the reviewer's suggestions. The “sulfides” has been added into page 2 line 43 for consistency with Ref. 16.

Comment 5. *However, the adsorption of H^* onto Fe atoms strongly depends on their difference in Pauling electronegativity. The change in charge density induced by structural phase transition mainly occurs at surficial Fe atom, leading to an increase of difference in Pauling electronegativity between H^* and Fe from 0.23 (P62c phase) to 0.37 (P63/mmc phase). Therefore, the Gibbs free energy of hydrogen adsorption tends to zero (as shown in Fig. 4a), which is benefit for improving catalytic activity.*

The authors have introduced the concept of Pauling electronegativity at this point. I am not convinced by their explanation, as the electronegativity should increase with the strength of the binding energy. They report instead an opposite trend. I believe that lines 183-188 should better be removed, especially because the results on the charge transfer of the upper panel of Fig. 4b are not there any more.

Answer: Thanks for the reviewer's suggestions. The inappropriate descriptions in lines 183-188 have been removed.

Comment 6. *From the analysis about its crystal symmetry, the Fe1 and Fe2 of P62c phase surface are distributed at two sides of symmetric axis as shown in Figure R3-3(a). When the structural phase transits from P62c to P63/mmc, the Fe1 and Fe2 atoms are distorted to symmetric axis (P63/mmc, Figure R3-3b) from opposite direction (marked by black arrow), finally tend to syequilibrium position (marked by*

purple line). Therefore, the Gibbs free energy of adsorption at Fe1 and Fe2 sites demonstrates opposite signs.

I do not agree with the authors on this point. It is unlikely that a different coordinate along the surface as that reported in Figure R3-3(a) would result in opposite adsorption energies. The difference between Fe1 and Fe2 may instead be due to their different positions along the normal to the surface, which is not possible to appreciate either from figure R3-3 or figure 4b. Can the authors look into it?

Answer: Thanks for the reviewer's kind reminding. After checking carefully our calculation process, we found that the positive values of free energy of Fe1 site were mistaken for negative ones due to our carelessness. Now, the recalculations indicate that the free energy of Fe1 site displays similar behavior to Fe2, as shown in Fig. R3-1, which is consistent with reviewer 3's speculation. This result has been corrected.

Figure R3-1. The Gibbs free energy as a function of the fraction of the atomic distortion between the $P\bar{6}2_c$ (0%) and $P6_3/mmc$ (100%) phase.

Comment 7. The atomic density of state (DOS) analysis confirms that the highest

occupied orbit is not composed by p-states of S atoms but d-states of Fe atoms (lower panel in Fig. 4b), and the corresponding electronic structure transition from semiconductor to metal phase (see detailed orbital resolved DOSs in Figs. S15 and S16) will benefit for improving carrier transfer and HER performance.

Is it important for the main message of the work that the highest occupied orbitals are composed by d-states of Fe atoms? Once showing (line 99) in figure 2b (perhaps figure S16 could be also referenced in line 99) the appearance of a band gap, do the lower panel of figure 4b, figure S15 and lines 188-193) contribute to make stronger the manuscript? Personally I think that they can be removed.

Answer: Thanks for the reviewer's suggestion. The lower panel of Figure 4b and Figure S15 and the corresponding descriptions in lines 188-193 have been deleted.

Comment 8. *The calculated result demonstrates that the energy along Volmer-Tafel reaction pathway (marked by black V-T line) is larger than that along Volmer-Heyrovsky (marked by red V-H) process, indicating that Volmer-Heyrovsky mechanism has obvious advantage in HER process.*

I appreciate that the authors have now included the Volmer-Heyrovsky mechanism. From the new comparison, it is again difficult to say which of the two mechanisms will occur, as both profiles present high activation barriers. Given this circumstance, I suggest the authors to use Fig. 5 only to present a possible pathway of the mechanism suggested by the experimental data, i.e. the Volmer-Heyrovsky (the phase used in these calculations should be mentioned). Lines 241-263 should be modified accordingly. Finally, the sentence "Therefore, the Volmer-Heyrovsky mechanism plays

a critical role in improving HER performance." should be deleted as I do not think that it is currently adequately supported by Fig. 5.

Answer: Thanks for the reviewer's kind suggestion. The phase used in our calculation has been added. The inappropriate descriptions have been revised, and the sentence "Therefore, the Volmer-Heyrovsky mechanism plays a critical role in improving HER performance." has been deleted"

End

REVIEWERS' COMMENTS:

Reviewer #2 (Remarks to the Author):

The authors have satisfactorily answered my comments and the paper has gained much clarity and quality now.

Provided the comments of Referee #3 have been answered, the paper seems suitable for publication in Nature Communications without further change.

Marion Giraud

Reviewer #3 (Remarks to the Author):

I am happy with most of the points raised in the second round of review, but two points in the rebuttal letter still need to be clarified.

Comment 2

The authors still mention the FeS (100) surface of the P-62c phase. This is the surface used by them for the P63/mmc phase but not for the P-62c phase.

Line 436: Please change to something like "FeS (100) surface for the P63/mmc phase (bottom panel) and its corresponding surface in the P-62c phase (top panel)"

Line 34 of SI. Please change to something like "The (100) surface with Fe and S termination of the P63/mmc phase and its corresponding surface in the P-62c phase are"

Line 314 of SI. Please change to something like "The (100) surface of the P63/mmc phase and its corresponding surface in the P-62c phase."

Alternatively, figure out the surface of the P-62c phase corresponding to that of the P63/mmc (100) surface.

Comment 8

As mentioned in my previous comment 15, from the comparison in Figure 5 it is difficult to say which of the two mechanisms will occur, as both profiles present very high activation barriers.

Lines 241-242: "Generally, the energy along V-T reaction pathway (marked by black lines) is larger than that along V-H mechanism (marked by red line)." This sentence has very little meaning.

Line 203: "which is in accordance with our DFT calculations in Fig. 5." should be removed.

I suggest again the authors to use Figure 5 to present a possible pathway only of the mechanism suggested by the experimental data, i.e. the Volmer-Heyrovsky. Lines 239-257 should be modified accordingly, while lines 258-259 become superfluous.

"Photoinduced Semiconductor-Metal Transition in Ultrathin Troilite FeS Nanosheets to Trigger Efficient Hydrogen Evolution"

ID: NCOMMS-18-27543B

Response to the report of the reviewer 3

Comment 1. *I am happy with most of the points raised in the second round of review, but two points in the rebuttal letter still need to be clarified.*

Question: Comment 2 The authors still mention the FeS (100) surface of the P-62c phase. This is the surface used by them for the P63/mmc phase but not for the P-62c phase.

Line 436: Please change to something like "FeS (100) surface for the P63/mmc phase (bottom panel) and its corresponding surface in the P-62c phase (top panel)"

Line 34 of SI. Please change to something like "The (100) surface with Fe and S termination of the P63/mmc phase and its corresponding surface in the P-62c phase are".

Line 314 of SI. Please change to something like "The (100) surface of the P63/mmc phase and its corresponding surface in the P-62c phase."

Alternatively, figure out the surface of the P-62c phase corresponding to that of the P63/mmc (100) surface.

Answer: Thanks for the reviewer's comments and suggestions. Those unsuitable description have been corrected and marked by red.

Comment 2. *Question: As mentioned in my previous comment 15, from the comparison in Figure 5 it is difficult to say which of the two mechanisms will occur, as both profiles present very high activation barriers.*

Lines 241-242: "Generally, the energy along V-T reaction pathway (marked by black lines) is larger than that along V-H mechanism (marked by red line)." This sentence has very little meaning. Line 203: "which is in accordance with our DFT calculations in Fig. 5." should be removed. I suggest again the authors to use Figure 5 to present a possible pathway only of the mechanism suggested by the experimental data, i.e. the Volmer-Heyrovsky. Lines 239-257 should be modified accordingly, while lines 258-259 become superfluous.

Answer: Thanks for the reviewer's comments and suggestions. Those unsuitable description have been removed.